# Convolutional Conditional Neural Processes

**Jonathan Gordon**[*]
University of Cambridge
jg801@cam.ac.uk

**Wessel P. Bruinsma**[*]
University of Cambridge
Invenia Labs
wpb23@cam.ac.uk

**Andrew Y. K. Foong**
University of Cambridge
ykf21@cam.ac.uk

**James Requeima**
University of Cambridge
Invenia Labs
jrr41@cam.ac.uk

**Yann Dubois**
University of Cambridge
yanndubois96@gmail.com

**Richard E. Turner**
University of Cambridge
Microsoft Research
ret26@cam.ac.uk

## ABSTRACT

We introduce the Convolutional Conditional Neural Process (CONVCNP), a new member of the Neural Process family that models *translation equivariance* in the data. Translation equivariance is an important inductive bias for many learning problems including time series modelling, spatial data, and images. The model embeds data sets into an infinite-dimensional function space as opposed to a finite-dimensional vector space. To formalize this notion, we extend the theory of neural representations of sets to include *functional representations*, and demonstrate that any translation-equivariant embedding can be represented using a *convolutional deep set*. We evaluate CONVCNPs in several settings, demonstrating that they achieve state-of-the-art performance compared to existing NPs. We demonstrate that building in translation equivariance enables zero-shot generalization to challenging, out-of-domain tasks.

## 1 INTRODUCTION

Neural Processes (NPs; Garnelo et al., 2018b;a) are a rich class of models that define a conditional distribution $p(\boldsymbol{y}|\boldsymbol{x}, Z, \boldsymbol{\theta})$ over output variables $\boldsymbol{y}$ given input variables $\boldsymbol{x}$, parameters $\boldsymbol{\theta}$, and a set of observed data points in a *context set* $Z = \{\boldsymbol{x}_m, \boldsymbol{y}_m\}_{m=1}^{M}$. A key component of NPs is the embedding of context sets $Z$ into a representation space through an encoder $Z \mapsto E(Z)$, which is achieved using a DEEPSETS function approximator (Zaheer et al., 2017). This simple model specification allows NPs to be used for (i) meta-learning (Thrun & Pratt, 2012; Schmidhuber, 1987), since predictions can be generated on the fly from new context sets at test time; and (ii) multi-task or transfer learning (Requeima et al., 2019), since they provide a natural way of sharing information between data sets. Moreover, conditional NPs (CNPs; Garnelo et al., 2018a), a deterministic variant of NPs, can be trained in a particularly simple way with maximum likelihood learning of the parameters $\boldsymbol{\theta}$, which mimics how the system is used at test time, leading to strong performance (Gordon et al., 2019).

Natural application areas of NPs include time series, spatial data, and images with missing values. Consequently, such domains have been used extensively to benchmark current NPs (Garnelo et al., 2018a;b; Kim et al., 2019). Often, ideal solutions to prediction problems in such domains should be translation equivariant: if the data are translated in time or space, then the predictions should be translated correspondingly (Kondor & Trivedi, 2018; Cohen & Welling, 2016). This relates to the notion of stationarity. As such, NPs would ideally have translation equivariance built directly into the modelling assumptions as an inductive bias. Unfortunately, current NP models must learn this structure from the data set instead, which is sample and parameter inefficient as well as impacting the ability of the models to generalize.

The goal of this paper is to build translation equivariance into NPs. Famously, convolutional neural networks (CNNs) added translation equivariance to standard multilayer perceptrons (LeCun et al., 1998; Cohen & Welling, 2016). However, it is not straightforward to generalize NPs in an analogous way: (i) CNNs require data to live "on the grid" (e.g. image pixels form a regularly spaced grid), while many of the above domains have data that live "off the grid" (e.g. time series data may be observed irregularly at any time $t \in \mathbb{R}$). (ii) NPs operate on partially observed context sets whereas

---

[*]Authors contributed equally. Complete description of author contributions in Appendix E

CNNs typically do not. (iii) NPs rely on embedding sets into a finite-dimensional vector space for which the notion of equivariance with respect to input translations is not natural, as we detail in Section 3. In this work, we introduce the CONVCNP, a new member of the NP family that accounts for translation equivariance.[1] This is achieved by extending the theory of learning on sets to include *functional representations*, which in turn can be used to express any translation-equivariant NP model. Our key contributions can be summarized as follows.

(i) We provide a representation theorem for translation-equivariant functions on sets, extending a key result of Zaheer et al. (2017) to *functional embeddings*, including sets of varying size.

(ii) We extend the NP family of models to include translation equivariance.

(iii) We evaluate the CONVCNP and demonstrate that it exhibits excellent performance on several synthetic and real-world benchmarks.

## 2 BACKGROUND AND FORMAL PROBLEM STATEMENT

In this section we introduce the notation and precisely define the problem this paper addresses.

**Notation.** In the following, let $\mathcal{X} = \mathbb{R}^d$ and $\mathcal{Y} \subseteq \mathbb{R}^{d'}$ (with $\mathcal{Y}$ compact) be the spaces of inputs and outputs respectively. To ease notation, we often assume scalar outputs $\mathcal{Y} \subseteq \mathbb{R}$. Define $\mathcal{Z}_M = (\mathcal{X} \times \mathcal{Y})^M$ as the collection of $M$ input–output pairs, $\mathcal{Z}_{\leq M} = \bigcup_{m=1}^{M} \mathcal{Z}_m$ as the collection of at most $M$ pairs, and $\mathcal{Z} = \bigcup_{m=1}^{\infty} \mathcal{Z}_m$ as the collection of finitely many pairs. Since we will consider *permutation-invariant* functions on $\mathcal{Z}$ (defined later in Property 1), we may refer to elements of $\mathcal{Z}$ as sets or data sets. Furthermore, we will use the notation $[n] = \{1, \ldots, n\}$.

**Conditional Neural Processes (CNPs).** CNPs model predictive distributions as $p(\boldsymbol{y}|\boldsymbol{x}, Z) = p(\boldsymbol{y}|\Phi(\boldsymbol{x}, Z), \boldsymbol{\theta})$, where $\Phi$ is defined as a composition $\rho \circ E$ of an encoder $E \colon \mathcal{Z} \to \mathbb{R}^e$ mapping into the *embedding space* $\mathbb{R}^e$ and a decoder $\rho \colon \mathbb{R}^e \to C_b(\mathcal{X}, \mathcal{Y})$. Here $E(Z) \in \mathbb{R}^e$ is a *vector representation* of the set $Z$, and $C_b(\mathcal{X}, \mathcal{Y})$ is the space of continuous, bounded functions $\mathcal{X} \to \mathcal{Y}$ endowed with the supremum norm. While NPs (Garnelo et al., 2018b) employ latent variables to indirectly specify predictive distributions, in this work we focus on CNP models which do not.

As noted by Lee et al. (2019); Bloem-Reddy & Teh (2019), since $E$ is a function on sets, the form of $\Phi$ in CNPs tightly relates to the growing literature on learning and representing functions on sets (Zaheer et al., 2017; Qi et al., 2017a; Wagstaff et al., 2019). Central to this body of work is the notion that, because the elements of a set have no order, functions on sets are naturally permutation invariant. Hence, to view functions on $\mathcal{Z}$ as functions on sets, we require such functions to be permutation invariant. This notion is formalized in Property 1.

**Property 1** ($\mathbb{S}_n$-invariant and $\mathbb{S}$-invariant functions). Let $\mathbb{S}_n$ be the group of permutations of $n$ symbols for $n \in \mathbb{N}$. A function $\Phi$ on $\mathcal{Z}_n$ is called $\mathbb{S}_n$-*invariant* if
$$\Phi(Z_n) = \Phi(\pi Z_n) \quad \text{for all } \pi \in \mathbb{S}_n \text{ and } Z_n \in \mathcal{Z}_n,$$
where the application of $\pi$ to $Z_n$ is defined as $\pi Z_n = ((\boldsymbol{x}_{\pi(1)}, \boldsymbol{y}_{\pi(1)}), \ldots, (\boldsymbol{x}_{\pi(n)}, \boldsymbol{y}_{\pi(n)}))$. A function $\Phi$ on $\mathcal{Z}$ is called $\mathbb{S}$-*invariant* if the restrictions $\Phi|_{\mathcal{Z}_n}$ are $\mathbb{S}_n$-*invariant* for all $n$.

Zaheer et al. (2017) demonstrate that any continuous $\mathbb{S}_M$-invariant function $f \colon \mathcal{Z}_M \to \mathbb{R}$ has a *sum-decomposition* (Wagstaff et al., 2019), i.e. a representation of the form $f(Z) = \rho(\sum_{\boldsymbol{z} \in Z} \phi(\boldsymbol{z}))$ for appropriate $\rho$ and $\phi$ (though this could only be shown for fixed-sized sets). This is indeed the form employed by the NP family for the encoder that embeds sets into a latent representation.

**Translation equivariance.** The focus of this work is on models that are *translation equivariant*: if the input locations of the data are translated by an amount $\boldsymbol{\tau}$, then the predictions should be translated correspondingly. Translation equivariance for functions operating on sets is formalized in Property 2.

**Property 2** (Translation equivariant mappings on sets). Let $\mathcal{H}$ be an appropriate space of functions on $\mathcal{X}$, and define $T$ and $T'$ as follows:
$$T \colon \mathcal{X} \times \mathcal{Z} \to \mathcal{Z}, \qquad T_{\boldsymbol{\tau}} Z = ((\boldsymbol{x}_1 + \boldsymbol{\tau}, \boldsymbol{y}_1), \ldots, (\boldsymbol{x}_m + \boldsymbol{\tau}, \boldsymbol{y}_m)),$$
$$T' \colon \mathcal{X} \times \mathcal{H} \to \mathcal{H}, \qquad T'_{\boldsymbol{\tau}} h(\boldsymbol{x}) = h(\boldsymbol{x} - \boldsymbol{\tau}).$$

---

[1]Source code available at `https://github.com/cambridge-mlg/convcnp`.

Then a mapping $\Phi\colon \mathcal{Z} \to \mathcal{H}$ is called *translation equivariant* if $\Phi(T_{\boldsymbol{\tau}} Z) = T'_{\boldsymbol{\tau}} \Phi(Z)$ for all $\boldsymbol{\tau} \in \mathcal{X}$ and $Z \in \mathcal{Z}$.

Having formalized the problem, we now describe how to construct CNPs that translation equivariant.

## 3 CONVOLUTIONAL DEEP SETS

We are interested in translation equivariance (Property 2) with respect to translations on $\mathcal{X}$. The NP family encoder maps sets $Z$ to an embedding in a vector space $\mathbb{R}^d$, for which the notion of equivariance with respect to input translations in $\mathcal{X}$ is not well defined. For example, a function $f$ on $\mathcal{X}$ can be translated by $\boldsymbol{\tau} \in \mathcal{X}$: $f(\,\cdot\, - \boldsymbol{\tau})$. However, for a vector $\boldsymbol{x} \in \mathbb{R}^d$, which can be seen as a function $[d] \to \mathbb{R}$, $\boldsymbol{x}(i) = x_i$, the translation $\boldsymbol{x}(\,\cdot\, - \boldsymbol{\tau})$ is not well-defined. To overcome this issue, we enrich the encoder $E\colon \mathcal{Z} \to \mathcal{H}$ to map into a *function space* $\mathcal{H}$ containing functions on $\mathcal{X}$. Since functions in $\mathcal{H}$ map from $\mathcal{X}$, our notion of translation equivariance (Property 2) is now also well defined for $E(Z)$. As we demonstrate below, every translation-equivariant function on sets has a representation in terms of a specific functional embedding.

**Definition 1** (Functional mappings on sets and functional representations of sets). Call a map $E\colon \mathcal{Z} \to \mathcal{H}$ a *functional mapping on sets* if it maps from sets $\mathcal{Z}$ to an appropriate space of functions $\mathcal{H}$. Furthermore, call $E(Z)$ the *functional representation* of the set $Z$.

Considering functional representations of sets leads to the key result of this work, which can be summarized as follows. For $\mathcal{Z}' \subseteq \mathcal{Z}$ appropriate, a continuous function $\Phi\colon \mathcal{Z}' \to C_b(\mathcal{X}, \mathcal{Y})$ satisfies Properties 1 and 2 if and only if it has a representation of the form

$$\Phi(Z) = \rho\left(E(Z)\right), \quad E(Z) = \sum\nolimits_{(\boldsymbol{x}, \boldsymbol{y}) \in Z} \phi(\boldsymbol{y})\psi(\,\cdot\, - \boldsymbol{x}) \in \mathcal{H}, \tag{1}$$

for some continuous and translation-equivariant $\rho\colon \mathcal{H} \to C_b(\mathcal{X}, \mathcal{Y})$, and appropriate $\phi$ and $\psi$. Note that $\rho$ is a map between function spaces. We also remark that continuity of $\Phi$ is not in the usual sense; we return to this below.

Equation (1) defines the encoder used by our proposed model, the CONVCNP. In Section 3.1, we present our theoretical results in more detail. In particular, Theorem 1 establishes equivalence between any function satisfying Properties 1 and 2 and the representational form in Equation (1). In doing so, we provide an extension of the key result of Zaheer et al. (2017) to functional representations on sets, and show that it can naturally be extended to handle varying-size sets. The practical implementation of CONVCNPs — the design of $\rho$, $\phi$, and $\psi$ — is informed by our results in Section 3.1 (as well as the proofs, provided in Appendix A), and is discussed for domains of interest in Section 4.

### 3.1 REPRESENTATIONS OF TRANSLATION EQUIVARIANT FUNCTIONS ON SETS

In this section we establish the theoretical foundation of the CONVCNP. We begin by stating a definition that is used in our main result. We denote $[m] = \{1, \ldots, m\}$.

**Definition 2** (Multiplicity). A collection $\mathcal{Z}' \subseteq \mathcal{Z}$ is said to have *multiplicity* $K$ if, for every set $Z \in \mathcal{Z}'$, every $\boldsymbol{x}$ occurs at most $K$ times:

$$\operatorname{mult} \mathcal{Z}' := \sup\left\{\sup\left\{\underbrace{|\{i \in [m] : \boldsymbol{x}_i = \hat{\boldsymbol{x}}\}|}_{\text{number of times every } \boldsymbol{x} \text{ occurs}} : \hat{\boldsymbol{x}} = \boldsymbol{x}_1, \ldots, \boldsymbol{x}_m\right\} : (\boldsymbol{x}_i, y_i)_{i=1}^m \in \mathcal{Z}'\right\} = K.$$

For example, in the case of real-world data like time series and images, we often observe only one (possibly multi-dimensional) observation per input location, which corresponds to multiplicity one. We are now ready to state our key theorem.

**Theorem 1.** *Consider an appropriate[2] collection $\mathcal{Z}'_{\leq M} \subseteq \mathcal{Z}_{\leq M}$ with multiplicity $K$. Then a function $\Phi\colon \mathcal{Z}'_{\leq M} \to C_b(\mathcal{X}, \mathcal{Y})$ is continuous[3], permutation invariant (Property 1), and translation*

---

[2]For every $m \in [M]$, $\mathcal{Z}'_{\leq M} \cap \mathcal{Z}_m$ must be topologically closed and closed under permutations and translations.

[3]For every $m \in [M]$, the restriction $\Phi|_{\mathcal{Z}'_{\leq M} \cap \mathcal{Z}_m}$ is continuous.

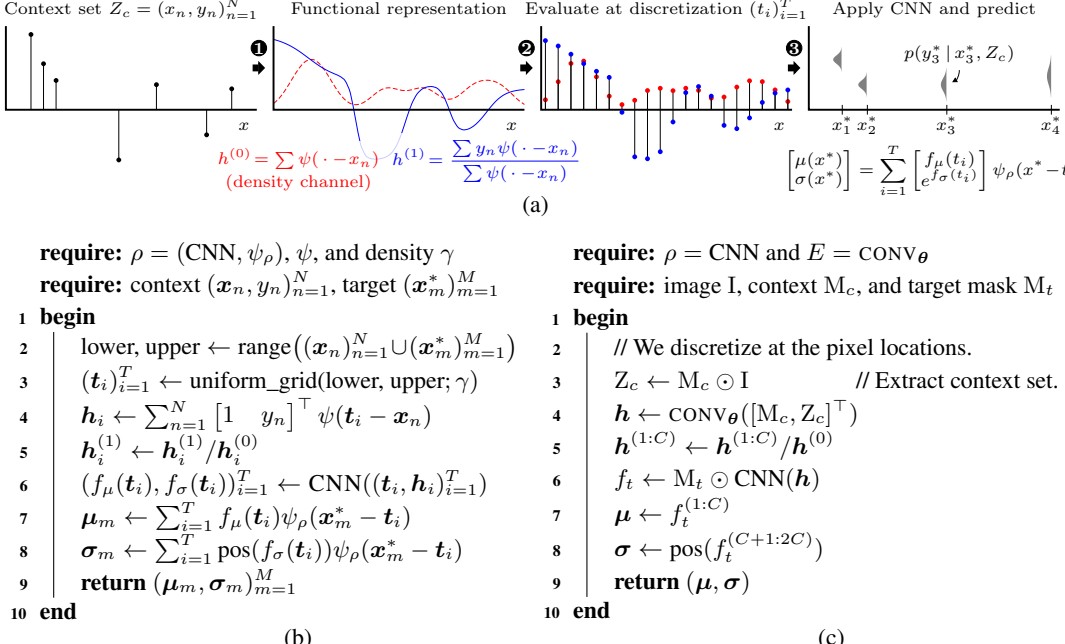

Figure 1: (a) Illustration of the CONVCNP forward pass in the off-the-grid case and pseudo-code for (b) off-the-grid and (c) on-the-grid data. The function $\text{pos} \colon \mathbb{R} \to (0, \infty)$ is used to enforce positivity.

*equivariant (Property 2) if and only if it has a representation of the form*

$$\Phi(Z) = \rho\left(E(Z)\right), \quad E((\boldsymbol{x}_1, y_1), \ldots, (\boldsymbol{x}_m, y_m)) = \sum_{i=1}^{m} \phi(y_i)\psi(\cdot - \boldsymbol{x}_i)$$

*for some continuous and translation-equivariant $\rho \colon \mathcal{H} \to C_b(\mathcal{X}, \mathcal{Y})$ and some continuous $\phi \colon \mathcal{Y} \to \mathbb{R}^{K+1}$ and $\psi \colon \mathcal{X} \to \mathbb{R}$, where $\mathcal{H}$ is an appropriate space of functions that includes the image of $E$. We call a function $\Phi$ of the above form* CONVDEEPSET.

The proof of Theorem 1 is provided in Appendix A. We here discuss several key points from the proof that have practical implications and provide insights for the design of CONVCNPs: (i) For the construction of $\rho$ and $E$, $\psi$ is set to a flexible positive-definite kernel associated with a Reproducing Kernel Hilbert Space (RKHS; Aronszajn (1950)), which results in desirable properties for $E$. (ii) Using the work by Zaheer et al. (2017), we set $\phi(y) = (y^0, y^1, \cdots, y^K)$ to be the powers of $y$ up to order $K$. (iii) Theorem 1 requires $\rho$ to be a powerful function approximator of continuous, translation-equivariant maps between functions. In Section 4, we discuss how these theoretical results inform our implementations of CONVCNPs.

Theorem 1 extends the result of Zaheer et al. (2017) discussed in Section 2 by embedding the set into an infinite-dimensional space — the RKHS — instead of a finite-dimensional space. Beyond allowing the model to exhibit translation equivariance, the RKHS formalism allows us to naturally deal with finite sets of varying sizes, which turns out to be challenging with finite-dimensional embeddings. Furthermore, our formalism requires $\phi(y) = (y^0, y^1, y^2, \ldots, y^K)$ to expand up to order no more than the *multiplicity* of the sets $K$; if $K$ is bounded, then our results hold for sets up to any arbitrarily large finite size $M$, while fixing $\phi$ to be only $(K + 1)$-dimensional.

## 4 CONVOLUTIONAL CONDITIONAL NEURAL PROCESSES

In this section we discuss the architectures and implementation details for CONVCNPs. Similar to NPs, CONVCNPs model the conditional distribution as

$$p(\boldsymbol{Y}|\boldsymbol{X}, Z) = \prod_{n=1}^{N} p(\boldsymbol{y}_n|\Phi_{\boldsymbol{\theta}}(Z)(\boldsymbol{x}_n)) = \prod_{n=1}^{N} \mathcal{N}(\boldsymbol{y}_n; \boldsymbol{\mu}_n, \boldsymbol{\Sigma}_n) \text{ with } (\boldsymbol{\mu}_n, \boldsymbol{\Sigma}_n) = \Phi_{\boldsymbol{\theta}}(Z)(\boldsymbol{x}_n), \tag{2}$$

where $Z$ is the observed data and $\Phi$ a CONVDEEPSET. The key considerations are the design of $\rho$, $\phi$, and $\psi$ for $\Phi$. We provide separate models for data that lie on-the-grid and data that lie off-the-grid.

**Form of $\phi$.** The applications considered in this work have a single (potentially multi-dimensional) output per input location, so the multiplicity of $\mathcal{Z}$ is one (i.e., $K = 1$). It then suffices to let $\phi$ be a power series of order one, which is equivalent to appending a constant to $\boldsymbol{y}$ in all data sets, i.e. $\phi(\boldsymbol{y}) = [1 \; \boldsymbol{y}]^{\top}$. The first output $\phi_1$ thus provides the model with information regarding where data has been observed, which is necessary to distinguish between no observed datapoint at $\boldsymbol{x}$ and a datapoint at $\boldsymbol{x}$ with $\boldsymbol{y} = \boldsymbol{0}$. Denoting the functional representation as $\boldsymbol{h}$, we can think of the first channel $\boldsymbol{h}^{(0)}$ as a "density channel". We found it helpful to divide the remaining channels $\boldsymbol{h}^{(1:)}$ by $\boldsymbol{h}^{(0)}$ (Figure 1b, line 5), as this improved performance when there is large variation in the density of input locations. In the image processing literature, this is known as *normalized convolution* (Knutsson & Westin, 1993). The normalization operation can be reversed by $\rho$ and is therefore not restrictive.

**CONVCNPs for off-the-grid data.** Having specified $\phi$, it remains to specify the form of $\psi$ and $\rho$. Our proof of Theorem 1 suggests that $\psi$ should be a stationary, non-negative, positive-definite kernel. The exponentiated-quadratic (EQ) kernel with a learnable length scale parameter is a natural choice. This kernel is multiplied by $\phi$ to form the functional representation $E(Z)$ (Figure 1b, line 4; and Figure 1a, arrow 1).

Next, Theorem 1 suggests that $\rho$ should be a continuous, translation-equivariant map between function spaces. Kondor & Trivedi (2018) show that, in deep learning, any translation-equivariant model has a representation as a CNN. However, CNNs operate on discrete (on-the-grid) input spaces and produce discrete outputs. In order to approximate $\rho$ with a CNN, we discretize the input of $\rho$, apply the CNN, and finally transform the CNN output back to a continuous function $\mathcal{X} \to \mathcal{Y}$. To do this, for each context and test set, we space points $(\boldsymbol{t}_i)_{i=1}^{n} \subseteq \mathcal{X}$ on a uniform grid (at a pre-specified density) over a hyper-cube that covers both the context and target inputs. We then evaluate $(E(Z)(\boldsymbol{t}_i))_{i=1}^{n}$ (Figure 1b, lines 2–3; Figure 1a, arrow 2). This discretized representation of $E(Z)$ is then passed through a CNN (Figure 1b, line 6; Figure 1a, arrow 3).

To map the output of the CNN back to a continuous function $\mathcal{X} \to \mathcal{Y}$, we use the CNN outputs as weights for evenly-spaced basis functions (again employing the EQ kernel), which we denote by $\psi_{\rho}$ (Figure 1b, lines 7–8; Figure 1a, arrow 3). The resulting approximation to $\rho$ is not perfectly translation equivariant, but will be approximately so for length scales larger than the spacing of $(E(Z)(\boldsymbol{t}_i))_{i=1}^{n}$. The resulting continuous functions are then used to generate the (Gaussian) predictive mean and variance at any input. This, in turn, can be used to evaluate the log-likelihood.

**CONVCNP for on-the-grid data.** While CONVCNP is readily applicable to many settings where data live on a grid, in this work we focus on the image setting. As such, the following description uses the image completion task as an example, which is often used to benchmark NPs (Garnelo et al., 2018a; Kim et al., 2019). Compared to the off-the-grid case, the implementation becomes simpler as we can choose the discretization $(\boldsymbol{t}_i)_{i=1}^{n}$ to be the pixel locations.

Let $\mathrm{I} \in \mathbb{R}^{H \times W \times C}$ be an image — $H, W, C$ denote the height, width, and number of channels, respectively — and let $\mathrm{M}_c$ be the context mask, which is such that $[\mathrm{M}_c]_{i,j} = 1$ if pixel location $(i, j)$ is in the context set, and 0 otherwise. To implement $\phi$, we select all context points, $\mathrm{Z}_c := \mathrm{M}_c \odot \mathrm{I}$, and prepend the context mask: $\phi = [\mathrm{M}_c, \mathrm{Z}_c]^{\top}$ (Figure 1c, line 4).

Next, we apply a convolution to the context mask to form the density channel: $\boldsymbol{h}^{(0)} = \mathrm{CONV}_{\boldsymbol{\theta}}(\mathrm{M}_c)$ (Figure 1c, line 4). To all other channels, we apply a normalized convolution: $\boldsymbol{h}^{(1:C)} = \mathrm{CONV}_{\boldsymbol{\theta}}(\boldsymbol{y})/\boldsymbol{h}^{(0)}$ (Figure 1c, line 5), where the division is element-wise. The filter of the convolution is analogous to $\psi$, which means that $\boldsymbol{h}$ is the functional representation, with the convolution performing the role of $E$ (the summation in Figure 1b, line 4). Although the theory suggests using a non-negative, positive-definite kernel, we did not find significant empirical differences between an EQ kernel and using a fully trainable kernel restricted to positive values to enforce non-negativity (see Appendices D.4 and D.5 for details).

Lastly, we describe the on-the-grid version of $\rho(\cdot)$, which consists of two stages. First, we apply a CNN to $E(Z)$ (Figure 1c, line 6). Second, we apply a shared, pointwise MLP that maps the output of the CNN at each pixel location in the target set to $\mathbb{R}^{2C}$, where we absorb MLP into the CNN (MLP can be viewed as an 1×1 convolution). The first $C$ outputs are the means of a Gaussian predictive distribution and the second $C$ the standard deviations, which then pass through a positivity-enforcing

function (Figure 1c, line 7–8). To summarise, the on-the-grid algorithm is given by

$$(\boldsymbol{\mu}, \mathrm{pos}^{-1}(\boldsymbol{\sigma})) = \mathrm{CNN}(\underbrace{[\underbrace{\mathrm{CONV}(\mathrm{M}_c)}_{\text{density channel}}; \underbrace{\mathrm{CONV}(\mathrm{M}_c \odot \mathrm{I})/\,\mathrm{CONV}(\mathrm{M}_c)}_{\text{multiplies by } \psi \text{ and sums}}]^\top}_{\rho}), \qquad (3)$$

where $(\boldsymbol{\mu}, \boldsymbol{\sigma})$ are the image mean and standard deviation, $\rho$ is implemented with CNN, and $E$ is implemented with the mask $\mathrm{M}_c$ and convolution CONV.

**Training.** Denoting the data set $D = \{Z_n\}_{n=1}^N \subseteq \mathcal{Z}$ and the parameters by $\boldsymbol{\theta}$, maximum-likelihood training involves (Garnelo et al., 2018a;b)

$$\boldsymbol{\theta}^* = \arg\max_{\boldsymbol{\theta} \in \Theta} \sum_{n=1}^N \sum_{(\boldsymbol{x}, \boldsymbol{y}) \in Z_{n,t}} \log p(\boldsymbol{y} \,|\, \Phi_{\boldsymbol{\theta}}(Z_{n,c})(\boldsymbol{x})), \qquad (4)$$

where we have split $Z_n$ into context ($Z_{n,c}$) and target ($Z_{n,t}$) sets. This is standard practice in the NP (Garnelo et al., 2018a;b) and meta-learning settings (Finn et al., 2017; Gordon et al., 2019) and relates to neural auto-regressive models (Requeima et al., 2019). Note that the context set and target set are disjoint ($Z_{n,c} \cap Z_{n,t} = \emptyset$), which differs from the protocol for the NP (Garnelo et al., 2018a). Practically, stochastic gradient descent methods (Bottou, 2010) can be used for optimization.

## 5 EXPERIMENTS AND RESULTS

We evaluate the performance of CONVCNPs in both on-the-grid and off-the-grid settings focusing on two central questions: (i) Do translation-equivariant models improve performance in appropriate domains? (ii) Can translation equivariance enable CONVCNPs to generalize to settings outside of those encountered during training? We use several off-the-grid data-sets which are irregularly sampled time series ($\mathcal{X} = \mathbb{R}$), comparing to Gaussian processes (GPs; Williams & Rasmussen (2006)) and ATTNCNP(which is identical to the ANP (Kim et al., 2019), but without the latent path in the encoder), the best performing member of the CNP family. We then evaluate on several on-the-grid image data sets ($\mathcal{X} = \mathbb{Z}^2$). In all settings we demonstrate substantial improvements over existing neural process models. For the CNN component of our model, we propose a small and large architecture for each experiment (in the experimental sections named CONVCNP and CONVCNPXL, respectively). We note that these architectures are different for off-the-grid and on-the-grid experiments, with full details regarding the architectures given in the appendices.

### 5.1 SYNTHETIC 1D EXPERIMENTS

First we consider synthetic regression problems. At each iteration, a function is sampled, followed by context and target sets. Beyond EQ-kernel GPs (as proposed in Garnelo et al. (2018a); Kim et al. (2019)), we consider more complex data arising from Matern–$\frac{5}{2}$ and weakly-periodic kernels, as well as a challenging, non-Gaussian sawtooth process with random shift and frequency (see Figure 2, for example). CONVCNP is compared to CNP (Garnelo et al., 2018a) and ATTNCNP. Training and testing procedures are fixed across all models. Full details on models, data generation, and training procedures are provided in Appendix C.

Table 1: Log-likelihood from synthetic 1-dimensional experiments.

| Model | Params | EQ | Weak Periodic | Matern | Sawtooth |
|---|---|---|---|---|---|
| CNP | 66818 | -0.86 ± 3e-3 | -1.23 ± 2e-3 | -0.95 ± 1e-3 | -0.16 ± 1e-5 |
| ATTNCNP | 149250 | 0.72 ± 4e-3 | -1.20 ± 2e-3 | 0.10 ± 2e-3 | -0.16 ± 2e-3 |
| CONVCNP | 6537 | 0.70 ± 5e-3 | -0.92 ± 2e-3 | 0.32 ± 4e-3 | 1.43 ± 4e-3 |
| CONVCNPXL | 50617 | **1.06 ± 4e-3** | **-0.65 ± 2e-3** | **0.53 ± 4e-3** | **1.94 ± 1e-3** |

Table 1 reports the log-likelihood means and standard errors of the models over 1000 tasks. The context and target points for both training and testing lie within the interval $[-2, 2]$ where training data was observed (marked "training data range" in Figure 2). Table 1 demonstrates that, even when extrapolation is not required, CONVCNP significantly outperforms other models in all cases, despite having fewer parameters.

Figure 2 demonstrates that CONVCNP generates excellent fits, even for challenging functions such as from the Matern–$\frac{5}{2}$ kernel and sawtooth. Moreover, Figure 2 compares the performance

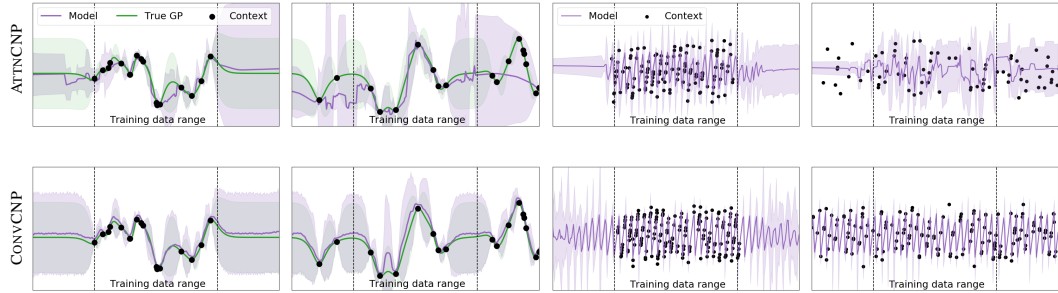

Figure 2: Example functions learned by the ATTNCNP (top row), and CONVCNP (bottom row), when trained on a Matern–$\frac{5}{2}$ kernel with length scale 0.25 (first and second column) and sawtooth function (third and fourth column). Columns one and three show the predictive posterior of the models when data is presented in same range as training, with predictive posteriors continuing beyond that range on either side. Columns two and four show model predictive posteriors when presented with data outside the training data range. Plots show means and two standard deviations.

of CONVCNP and ATTNCNP when data is observed outside the range where the models were trained: translation equivariance enables CONVCNP to elegantly generalize to this setting, whereas ATTNCNP is unable to generate reasonable predictions.

## 5.2 PLAsTiCC Experiments

The PLAsTiCC data set (Allam Jr et al., 2018) is a simulation of transients observed by the LSST telescope under realistic observational conditions. The data set contains 3,500,734 "light curves", where each measurement is of an object's brightness as a function of time, taken by measuring the photon flux in six different astronomical filters. The data can be treated as a six-dimensional time series. The data set was introduced in a Kaggle competition,[4] where the task was to use these light curves to classify the variable sources. The winning entry (Avocado, Boone, 2019) modeled the light curves with GPs and used these models to generate features for a gradient boosted decision tree classifier. We compare a multi-input–multi-output CONVCNP with the GP models used in Avocado.[5] CONVCNP accepts six channels as inputs, one for each astronomical filter, and returns 12 outputs — the means and standard deviations of six Gaussians. Full experimental details are given in Appendix C.3. The mean squared error of both approaches is similar, but the held-out log-likelihood from the CONVCNP is far higher (see Table 2).

Table 2: Mean and standard errors of log-likelihood and root mean squared error over 1000 test objects from the PLastiCC dataset.

| Model | Log-likelihood | MSE |
|---|---|---|
| Kaggle GP (Boone, 2019) | -0.335 $\pm$ 0.09 | 0.037 $\pm$ 4e-3 |
| ConvCP (ours) | **1.31** $\pm$ 0.30 | 0.040 $\pm$ 5e-3 |

## 5.3 Predator-Prey Models: Sim2Real

The CONVCNP model is well suited for applications where simulation data is plentiful, but real world training data is scarce (Sim2Real). The CONVCNP can be trained on a large amount of simulation data and then be deployed with real-world training data as the context set. We consider the Lotka–Volterra model (Wilkinson, 2011), which is used to describe the evolution of predator–prey populations. This model has been used in the Approximate Bayesian Computation literature where the task is to infer the parameters from samples drawn from the Lotka–Volterra process (Papamakarios & Murray, 2016). These methods do not simply extend to prediction problems such as interpolation or forecasting. In contrast, we train CONVCNP on synthetic data sampled from the Lotka–Volterra

---

[4]https://www.kaggle.com/c/PLAsTiCC-2018

[5]Full code for Avocado, including GP models, is available at https://github.com/kboone/avocado.

model and can then condition on real-world data from the Hudson's Bay lynx–hare data set (Leigh, 1968) to perform interpolation (see Figure 3; full experimental details are given in Appendix C.4). The CONVCNP performs accurate interpolation as shown in Figure 3. We were unable to successfully train the ATTNCNP for this task. We suspect this is because the simulation data are variable length-time series, which requires models to leverage translation equivariance at training time. As shown in Section 5.1, the ATTNCNP struggles to do this (see Appendix C.4 for complete details).

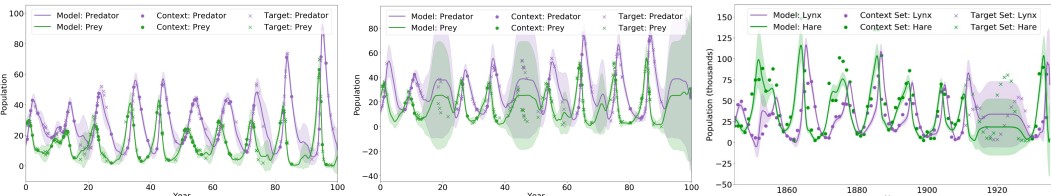

Figure 3: Left and centre: two samples from the Lotka–Volterra process (sim). Right: CONVCNP trained on simulations and applied to the Hudson's Bay lynx-hare dataset (real). Plots show means and two standard deviations.

## 5.4 2D IMAGE COMPLETION EXPERIMENTS

To test CONVCNP beyond one-dimensional features, we evaluate our model on on-the-grid image completion tasks and compare it to ATTNCNP. Image completion can be cast as a prediction of pixel intensities $y_i^*$ ($\in \mathbb{R}^3$ for RGB, $\in \mathbb{R}$ for greyscale) given a target 2D pixel location $x_i^*$ conditioned on an observed (context) set of pixel values $Z = (x_n, y_n)_{n=1}^N$. In the following experiments, the context set can vary but the target set contains all pixels from the image. Further experimental details are in Appendix D.1.

Table 3: Log-likelihood from image experiments (6 runs).

| Model | Params | MNIST | SVHN | CelebA32 | CelebA64 | ZSMM |
|---|---|---|---|---|---|---|
| ATTNCNP | 410k | 1.08 ±0.04 | 3.94 ±0.02 | 3.18 ±0.02 | | -0.83 ±0.08 |
| CONVCNP | 113k | 1.21 ±0.00 | 3.89 ±0.01 | 3.22 ±0.02 | 3.66 ±0.01 | **1.18 ±0.04** |
| CONVCNPXL | 400k | **1.27 ±0.01** | **3.97 ±0.02** | **3.39 ±0.02** | **3.73 ±0.01** | 0.86 ±0.12 |

**Standard benchmarks.** We first evaluate the model on four common benchmarks: MNIST (LeCun et al., 1998), SVHN (Netzer et al., 2011), and $32 \times 32$ and $64 \times 64$ CelebA (Liu et al., 2018). Importantly, these data sets are biased towards images containing a single, well-centered object. As a result, perfect translation-equivariance might hinder the performance of the model when the test data are similarly structured. We therefore also evaluated a larger CONVCNP that can learn such non-stationarity, while still sharing parameters across the input space (CONVCNPXL).

Table 3 shows that CONVCNP significantly outperforms ATTNCNP when it has a large receptive field size, while being at least as good with a small receptive field size. Qualitative samples for various context sets can be seen in Figure 5. Further qualitative comparisons and ablation studies can be found in Appendix D.3 and Appendix D.4 respectively.

**Generalization to multiple, non-centered objects.** The data sets from the previous paragraphs were centered and contained single objects. Here we test whether CONVCNPs trained on such data can generalize to images containing multiple, non-centered objects.

The last column of Table 3 evaluates the models in a zero shot multi-MNIST (ZSMM) setting, where images contain multiple digits at test time (Appendix D.2). CONVCNP significantly outperforms ATTNCNP on such tasks. Figure 4a shows a histogram of the image log-likelihoods for CONVCNP and ATTNCNP, as well as qualitative results at different percentiles of the CONVCNP distribution. CONVCNP is able to extrapolate to this out-of-distribution test set, while ATTNCNP appears to model the bias of the training data and predict a centered "mean" digit independently of the context. Interestingly, CONVCNPXL does not perform as well on this task. In particular, we find that, as the receptive field becomes very large, performance on this task decreases. We hypothesize that this has

to do with behavior of the model at the edges of the image. CNNs with larger receptive fields — the region of input pixels that affect a particular output pixel — are able to model non-stationary behavior by looking at the distance from any pixel to the image boundary. We expand on this discussion and provide further experimental evidence regarding the effects of receptive field on the ZSMM task in Appendix D.6.

Although ZSMM is a contrived task, note that our field of view usually contains multiple independent objects, thereby requiring translation equivariance. As a more realistic example, we took a CONVCNP model trained on CelebA and tested it on a natural image of different shape which contains multiple people (Figure 4b). Even with 95% of the pixels removed, the CONVCNP was able to produce a qualitatively reasonable reconstruction. A comparison with ATTNCNP is given in Appendix D.3.

**Computational efficiency.** Beyond the performance and generalization improvements, a key advantage of the CONVCNP is its computational efficiency. The memory and time complexity of a single self-attention layer grows quadratically with the number of inputs $M$ (the number of pixels for images) but only linearly for a convolutional layer.

Empirically, with a batch size of 16 on $32 \times 32$ MNIST, CONVCNPXL requires 945MB of VRAM, while ATTNCNP requires 5839 MB. For the $56 \times 56$ ZSMM CONVCNPXL increases its requirements to 1443 MB, while ATTNCNP could not fit onto a 32GB GPU. Ultimately, ATTNCNP had to be trained with a batch size of 6 (using 19139 MB) and we were not able to fit it for CelebA64. Recently, restricted attention has been proposed to overcome this computational issue (Parmar et al., 2018), but we leave an investigation of this and its relationship to CONVCNPs to future work.

## 6    RELATED WORK AND DISCUSSION

We have introduced CONVCNP, a new member of the CNP family that leverages embedding sets into function space to achieve translation equivariance. The relationship to (i) the NP family, and (ii) representing functions on sets, each imply extensions and avenues for future work.

**Deep sets.** Two key issues in the existing theory on learning with sets (Zaheer et al., 2017; Qi et al., 2017a; Wagstaff et al., 2019) are (i) the restriction to fixed-size sets, and (ii) that the dimensionality of the embedding space must be no less than the cardinality of the embedded sets. Our work implies that by considering appropriate embeddings into a function space, both issues are alleviated. In future work, we aim to further this analysis and formalize it in a more general context.

**Point-cloud models.** Another line of related research focuses on 3D point-cloud modelling (Qi et al., 2017a;b). While original work focused on permutation invariance (Qi et al., 2017a; Zaheer et al., 2017), more recent work has considered translation equivariance as well (Wu et al., 2019), leading to a model closely resembling CONVDEEPSETS. The key differences with our work are the following: (i) Wu et al. (2019) implement $\psi$ as an MLP with learned weights, resulting in a more flexible parameterization of the convolutional weights. (ii) Wu et al. (2019) interpret the computations as Monte Carlo approximations to an underlying continuous convolution, whereas we consider the problem of function approximation directly on sets. (iii) Wu et al. (2019) only consider the point-cloud application, whereas our derivation and modelling work considers general sets.

**Correlated samples and consistency under marginalization.** In the predictive distribution of CONVCNP (Equation (2)), predicted $\boldsymbol{y}$s are conditionally independent given the context set. Consequently, samples from the predictive distribution lack correlations and appear noisy. One solution is to instead define the predictive distribution in an autoregressive way, like e.g. PixelCNN++ (Salimans et al., 2017). Although samples are now correlated, the quality of the samples depends on the order in which the points are sampled. Moreover, the predicted $\boldsymbol{y}$s are then not consistent under marginalization (Garnelo et al., 2018b; Kim et al., 2019). Consistency under marginalization is more generally an issue for neural autoregressive models (Salimans et al., 2017; Parmar et al., 2018), although consistent variants have been devised (Louizos et al., 2019). To overcome the consistency issue for CONVCNP, exchangeable neural process models (e.g. Korshunova et al., 2018; Louizos et al., 2019) may provide an interesting avenue. Another way to introduce dependencies between $\boldsymbol{y}$s is to employ latent variables as is done in neural processes (Garnelo et al., 2018b). However, such an approach only achieves *conditional consistency*: given a context set, the predicted $\boldsymbol{y}$s will be dependent and consistent under marginalization, but this does not lead to a consistent joint model that also includes the context set itself.

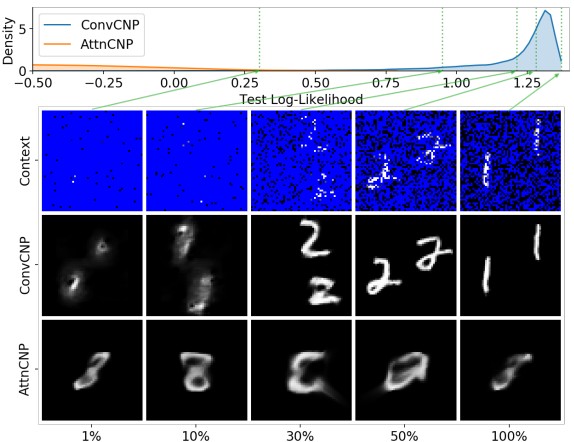 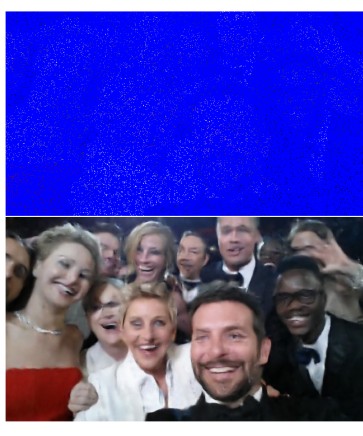

(a) Log-likelihood and qualitative results on ZSMM. The top row shows the log-likelihood distribution for both models. The images below correspond to the context points (top), CONVCNP target predictions (middle), and ATTNCNP target predictions (bottom). Each column corresponds to a given percentile of the CONVCNP distribution.

(b) Qualitative evaluation of a CON­VCNPXL trained on the unscaled CelebA ($218 \times 178$) and tested on Ellen's Oscar unscaled ($337 \times 599$) selfie (DeGeneres, 2014) with $5\%$ of the pixels as context (top).

Figure 4: Zero shot generalization to tasks that require translation equivariance.

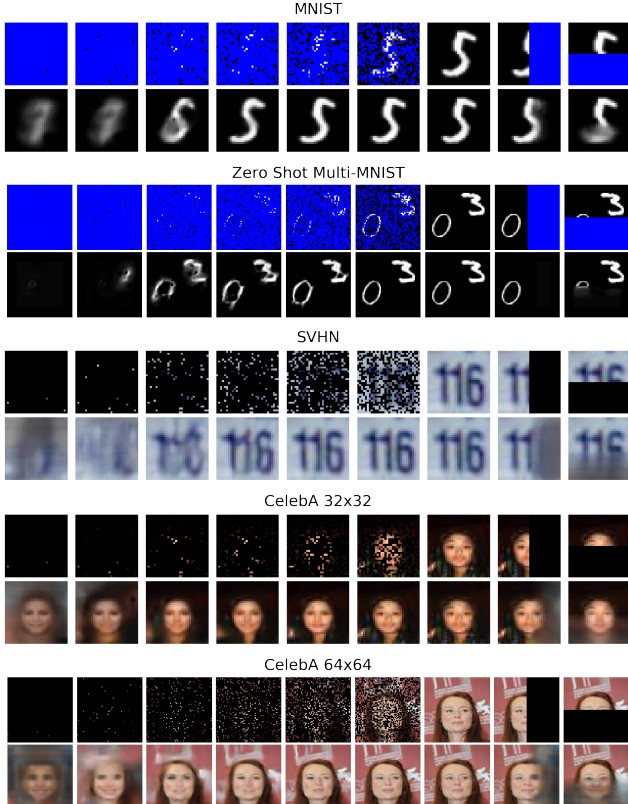

Figure 5: Qualitative evaluation of the CONVCNP(XL). For each dataset, an image is randomly sampled, the first row shows the given context points while the second is the mean of the estimated conditional distribution. From left to right the first seven columns correspond to a context set with 3, $1\%$, $5\%$, $10\%$, $20\%$, $30\%$, $50\%$, $100\%$ randomly sampled context points. In the last two columns, the context sets respectively contain all the pixels in the left and top half of the image. CONVCNPXL is shown for all datasets besides ZSMM, for which we show the fully translation equivariant CONVCNP.

ACKNOWLEDGEMENTS

We would like to thank Mark Rowland for help with checking the proofs, and David R. Burt, Will Tebbutt, Robert Pinsler, and Cozmin Ududec for helpful comments on the manuscript. Andrew Y. K. Foong is supported by a Trinity Hall Research Studentship and the George and Lilian Schiff Foundation. Richard E. Turner is supported by Google, Amazon, ARM, Improbable and EPSRC grants EP/M0269571 and EP/L000776/1.

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

# A    THEORETICAL RESULTS AND PROOFS

In this section, we provide the proof of Theorem 1. Our proof strategy is as follows. We first define an appropriate topology for fixed-sized sets (Appendix A.1). With this topology in place, we demonstrate that our proposed embedding into function space is homeomorphic (Lemmas 1 and 2). We then show that the embeddings of fixed-sized sets can be extended to varying-sized sets by "pasting" the embeddings together while maintaining their homeomorphic properties (Lemma 3). Following this, we demonstrate that the resulting embedding may be composed with a continuous mapping to our desired target space, resulting in a continuous mapping between two metric spaces (Lemma 4). Finally, in Appendix A.3 we combine the above-mentioned results to prove Theorem 1.

We begin with definitions that we will use throughout the section and then present our results. Let $\mathcal{X} = \mathbb{R}^d$ and let $\mathcal{Y} \subseteq \mathbb{R}$ be compact. Let $\psi$ be a symmetric, positive-definite kernel on $\mathcal{X}$. By the Moore–Aronszajn Theorem, there is a unique Hilbert space $(\mathcal{H}, \langle \cdot, \cdot \rangle_{\mathcal{H}})$ of real-valued functions on $\mathcal{X}$ for which $\psi$ is a reproducing kernel. This means that (i) $\psi(\cdot, \boldsymbol{x}) \in \mathcal{H}$ for all $\boldsymbol{x} \in \mathcal{X}$ and (ii) $\langle f, \psi(\cdot, \boldsymbol{x}) \rangle_{\mathcal{H}} = f(\boldsymbol{x})$ for all $f \in \mathcal{H}$ and $\boldsymbol{x} \in \mathcal{X}$ (reproducing property). For $\psi \colon \mathcal{X} \times \mathcal{X} \to \mathbb{R}$, $\boldsymbol{X} = (\boldsymbol{x}_1, \ldots, \boldsymbol{x}_n) \in \mathcal{X}^n$, and $\boldsymbol{X}' = (\boldsymbol{x}'_1, \ldots, \boldsymbol{x}'_n) \in \mathcal{X}^n$, we denote

$$\psi(\boldsymbol{X}, \boldsymbol{X}') = \begin{bmatrix} \psi(\boldsymbol{x}_1, \boldsymbol{x}'_1) & \cdots & \psi(\boldsymbol{x}_1, \boldsymbol{x}'_n) \\ \vdots & \ddots & \vdots \\ \psi(\boldsymbol{x}_n, \boldsymbol{x}'_1) & \cdots & \psi(\boldsymbol{x}_n, \boldsymbol{x}'_n) \end{bmatrix}.$$

**Definition 3** (Interpolating RKHS)**.** Call $\mathcal{H}$ *interpolating* if it interpolates any finite number of points: for every $((\boldsymbol{x}_i, y_i))_{i=1}^n \subseteq \mathcal{X} \times \mathcal{Y}$ with $(\boldsymbol{x}_i)_{i=1}^n$ all distinct, there is an $f \in \mathcal{H}$ such that $f(\boldsymbol{x}_1) = y_1, \ldots, f(\boldsymbol{x}_n) = y_n$.

For example, the RKHS induced by any strictly positive-definite kernel, e.g. the exponentiated quadratic (EQ) kernel $\psi(\boldsymbol{x}, \boldsymbol{x}') = \sigma^2 \exp(-\frac{1}{2\ell^2} \|\boldsymbol{x} - \boldsymbol{x}'\|^2)$, is interpolating: Let $\boldsymbol{c} = \psi(\boldsymbol{X}, \boldsymbol{X})^{-1} \boldsymbol{y}$ and consider $f = \sum_{i=1}^n c_i \psi(\cdot, \boldsymbol{x}_i) \in \mathcal{H}$. Then $f(\boldsymbol{X}) = \psi(\boldsymbol{X}, \boldsymbol{X}) \boldsymbol{c} = \boldsymbol{y}$.

## A.1    THE QUOTIENT SPACE $\mathcal{A}^n / \mathbb{S}_n$

Let $\mathcal{A}$ be a Banach space. For $\boldsymbol{x} = (x_1, \ldots, x_n) \in \mathcal{A}^n$ and $\boldsymbol{y} = (y_1, \ldots, y_n) \in \mathcal{A}^n$, let $\boldsymbol{x} \sim \boldsymbol{y}$ if $\boldsymbol{x}$ is a permutation of $\boldsymbol{y}$; that is, $\boldsymbol{x} \sim \boldsymbol{y}$ if and only if $\boldsymbol{x} = \pi \boldsymbol{y}$ for some $\pi \in \mathbb{S}_n$ where

$$\pi \boldsymbol{y} = (y_{\pi(1)}, \ldots, y_{\pi(n)}).$$

Let $\mathcal{A}^n / \mathbb{S}_n$ be the collection of equivalence classes of $\sim$. Denote the equivalence class of $\boldsymbol{x}$ by $[\boldsymbol{x}]$; for $A \subseteq \mathcal{A}^n$, denote $[A] = \{[\boldsymbol{a}] : \boldsymbol{a} \in A\}$. Call the map $\boldsymbol{x} \mapsto [\boldsymbol{x}] \colon \mathcal{A}^n \to \mathcal{A}^n / \mathbb{S}_n$ the canonical map. The natural topology on $\mathcal{A}^n / \mathbb{S}_n$ is the quotient topology, in which a subset of $\mathcal{A}^n / \mathbb{S}_n$ is open if and only if its preimage under the canonical map is open in $\mathcal{A}^n$. In what follows, we show that the quotient topology is metrizable.

On $\mathcal{A}^n$, since all norms on finite-dimensional vector spaces are equivalent, without loss of generality consider

$$\|\boldsymbol{x}\|_{\mathcal{A}^n}^2 = \sum_{i=1}^n \|x_i\|_{\mathcal{A}}^2.$$

Note that $\| \cdot \|_{\mathcal{A}^n}$ is permutation invariant: $\|\pi \cdot \|_{\mathcal{A}^n} = \| \cdot \|_{\mathcal{A}^n}$ for all $\pi \in \mathbb{S}_n$. On $\mathcal{A}^n / \mathbb{S}_n$, define

$$d \colon \mathcal{A}^n / \mathbb{S}_n \times \mathcal{A}^n / \mathbb{S}_n \to [0, \infty), \quad d([\boldsymbol{x}], [\boldsymbol{y}]) = \min_{\pi \in \mathbb{S}_n} \|\boldsymbol{x} - \pi \boldsymbol{y}\|_{\mathcal{A}^n}.$$

Call a set $[A] \subseteq \mathcal{A}^n / \mathbb{S}_n$ bounded if $\{d([\boldsymbol{x}], [0]) : [\boldsymbol{x}] \in [A]\}$ is bounded.

**Proposition 1.** *The function $d$ is a metric.*

*Proof.* We first show that $d$ is well defined on $\mathcal{A}^n / \mathbb{S}_n$. Assume $\boldsymbol{x} \sim \boldsymbol{x}'$ and $\boldsymbol{y} \sim \boldsymbol{y}'$. Then, $\boldsymbol{x}' = \pi_{\boldsymbol{x}} \boldsymbol{x}$ and $\boldsymbol{y}' = \pi_{\boldsymbol{y}} \boldsymbol{y}$. Using the group properties of $\mathbb{S}_n$ and the permutation invariance of $\| \cdot \|_{\mathcal{A}^n}$:

$$\begin{aligned} d([\boldsymbol{x}'], [\boldsymbol{y}']) &= \min_{\pi \in \mathbb{S}_n} \|\pi_{\boldsymbol{x}} \boldsymbol{x} - \pi \pi_{\boldsymbol{y}} \boldsymbol{y}\|_{\mathcal{A}^n} \\ &= \min_{\pi \in \mathbb{S}_n} \|\pi_{\boldsymbol{x}} \boldsymbol{x} - \pi \boldsymbol{y}\|_{\mathcal{A}^n} \\ &= \min_{\pi \in \mathbb{S}_n} \|\boldsymbol{x} - \pi_{\boldsymbol{x}}^{-1} \pi \boldsymbol{y}\|_{\mathcal{A}^n} \\ &= \min_{\pi \in \mathbb{S}_n} \|\boldsymbol{x} - \pi \boldsymbol{y}\|_{\mathcal{A}^n} \\ &= d([\boldsymbol{x}], [\boldsymbol{y}]). \end{aligned}$$

It is clear that $d([\boldsymbol{x}], [\boldsymbol{y}]) = d([\boldsymbol{y}], [\boldsymbol{x}])$ and that $d([\boldsymbol{x}], [\boldsymbol{y}]) = 0$ if and only if $[\boldsymbol{x}] = [\boldsymbol{y}]$. To show the triangle inequality, note that

$$\|\boldsymbol{x} - \pi_1 \pi_2 \boldsymbol{y}\|_{\mathcal{A}^n} \leq \|\boldsymbol{x} - \pi_1 \boldsymbol{z}\|_{\mathcal{A}^n} + \|\pi_1 \boldsymbol{z} - \pi_1 \pi_2 \boldsymbol{y}\|_{\mathcal{A}^n} = \|\boldsymbol{x} - \pi_1 \boldsymbol{z}\|_{\mathcal{A}^n} + \|\boldsymbol{z} - \pi_2 \boldsymbol{y}\|_{\mathcal{A}^n},$$

using permutation invariance of $\|\cdot\|_{\mathcal{A}^n}$. Hence, taking the minimum over $\pi_1$,

$$d([\boldsymbol{x}], [\boldsymbol{y}]) \leq d([\boldsymbol{x}], [\boldsymbol{z}]) + \|\boldsymbol{z} - \pi_2 \boldsymbol{y}\|_{\mathcal{A}^n},$$

so taking the minimum over $\pi_2$ gives the triangle inequality for $d$. $\qquad\square$

**Proposition 2.** *The canonical map $\mathcal{A}^n \to \mathcal{A}^n / \mathbb{S}_n$ is continuous under the metric topology induced by $d$.*

*Proof.* Follows directly from $d([\boldsymbol{x}], [\boldsymbol{y}]) \leq \|\boldsymbol{x} - \boldsymbol{y}\|_{\mathcal{A}^n}$. $\qquad\square$

**Proposition 3.** *Let $A \subseteq \mathcal{A}^n$ be topologically closed and closed under permutations. Then $[A]$ is topologically closed in $\mathcal{A}^n / \mathbb{S}_n$ under the metric topology.*

*Proof.* Recall that a subset $[A]$ of a metric space is closed iff every limit point of $[A]$ is also in $[A]$. Consider a sequence $([\boldsymbol{a}_n])_{n=1}^{\infty} \subseteq [A]$ converging to some $[\boldsymbol{x}] \in \mathcal{A}^n / \mathbb{S}_n$. Then there are permutations $(\pi_n)_{n=1}^{\infty} \subseteq \mathbb{S}_n$ such that $\pi_n \boldsymbol{a}_n \to \boldsymbol{x}$. Here $\pi_n \boldsymbol{a}_n \in A$, because $A$ is closed under permutations. Thus $\boldsymbol{x} \in A$, as $A$ is also topologically closed. We conclude that $[\boldsymbol{x}] \in [A]$. $\qquad\square$

**Proposition 4.** *Let $A \subseteq \mathcal{A}^n$ be open. Then $[A]$ is open in $\mathcal{A}^n / \mathbb{S}_n$ under the metric topology. In other words, the canonical map is open under the metric topology.*

*Proof.* Let $[\boldsymbol{x}] \in [A]$. Because $A$ is open, there is some ball $B_\varepsilon(\boldsymbol{y})$ with $\varepsilon > 0$ and $\boldsymbol{y} \in A$ such that $\boldsymbol{x} \in B_\varepsilon(\boldsymbol{y}) \subseteq A$. Then $[\boldsymbol{x}] \in B_\varepsilon([\boldsymbol{y}])$, since $d([\boldsymbol{x}], [\boldsymbol{y}]) \leq \|\boldsymbol{x} - \boldsymbol{y}\|_{\mathcal{A}^n} < \varepsilon$, and we claim that $B_\varepsilon([\boldsymbol{y}]) \subseteq [A]$. Hence $[\boldsymbol{x}] \in B_\varepsilon([\boldsymbol{y}]) \subseteq [A]$, so $[A]$ is open.

To show the claim, let $[\boldsymbol{z}] \in B_\varepsilon([\boldsymbol{y}])$. Then $d(\pi \boldsymbol{z}, \boldsymbol{y}) < \varepsilon$ for some $\pi \in \mathbb{S}_n$. Hence $\pi \boldsymbol{z} \in B_\varepsilon(\boldsymbol{y}) \subseteq A$, so $\pi \boldsymbol{z} \in A$. Therefore, $[\boldsymbol{z}] = [\pi \boldsymbol{z}] \in [A]$. $\qquad\square$

**Proposition 5.** *The quotient topology on $\mathcal{A}^n / \mathbb{S}_n$ induced by the canonical map is metrizable with the metric $d$.*

*Proof.* Since the canonical map is surjective, there exists exactly one topology on $\mathcal{A}^n / \mathbb{S}_n$ relative to which the canonical map is a quotient map: the quotient topology (Munkres, 1974).

Let $p \colon \mathcal{A}^n \to \mathcal{A}^n / \mathbb{S}_n$ denote the canonical map. It remains to show that $p$ is a quotient map under the metric topology induced by $d$; that is, we show that $U \subset \mathcal{A}^n / \mathbb{S}_n$ is open in $\mathcal{A}^n / \mathbb{S}_n$ under the metric topology if and only if $p^{-1}(U)$ is open in $\mathcal{A}^n$.

Let $p^{-1}(U)$ be open in $\mathcal{A}^n$. We have that $U = p(p^{-1}(U))$, so $U$ is open in $\mathcal{A}^n / \mathbb{S}_n$ under the metric topology by Proposition 4. Conversely, if $U$ is open in $\mathcal{A}^n / \mathbb{S}_n$ under the metric topology, then $p^{-1}(U)$ is open in $\mathcal{A}^n$ by continuity of the canonical map under the metric topology. $\qquad\square$

## A.2 EMBEDDINGS OF SETS INTO AN RKHS

Whereas $\mathcal{A}$ previously denoted an arbitrary Banach space, in this section we specialize to $\mathcal{A} = \mathcal{X} \times \mathcal{Y}$. We denote an element in $\mathcal{A}$ by $(\boldsymbol{x}, y)$ and an element in $\mathcal{Z}_M = \mathcal{A}^M$ by $((\boldsymbol{x}_1, y_1), \ldots, (\boldsymbol{x}_M, y_M))$. Alternatively, we denote $((\boldsymbol{x}_1, y_1), \ldots, (\boldsymbol{x}_M, y_M))$ by $(\boldsymbol{X}, \boldsymbol{y})$ where $\boldsymbol{X} = (\boldsymbol{x}_1, \ldots, \boldsymbol{x}_M) \in \mathcal{X}^M$ and $\boldsymbol{y} = (y_1, \ldots, y_M) \in \mathcal{Y}^M$. We clarify that an element in $\mathcal{Z}_M = \mathcal{A}^M$ is permuted as follows: for $\pi \in \mathbb{S}_M$,

$$\pi(\boldsymbol{X}, \boldsymbol{y}) = \pi((\boldsymbol{x}_1, y_1), \ldots, (\boldsymbol{x}_M, y_M)) = ((\boldsymbol{x}_{\pi(1)}, y_{\pi(1)}), \ldots, (\boldsymbol{x}_{\pi(n)}, y_{\pi(n)})) = (\pi \boldsymbol{X}, \pi \boldsymbol{y}).$$

Note that permutation-invariant functions on $\mathcal{Z}_M$ are in correspondence to functions on the quotient space induced by the equivalence class of permutations, $\mathcal{Z}_M / \mathbb{S}_m$ The latter is a more natural representation.

Lemma 3 states that it is possible to homeomorphically embed sets into an RKHS. This result is key to proving our main result. Before proving Lemma 3, we provide several useful results. We begin by demonstrating that an embedding of sets of a fixed size into a RKHS is continuous and injective.

**Lemma 1.** *Consider a collection $\mathcal{Z}_M' \subseteq \mathcal{Z}_M$ that has multiplicity $K$. Set*

$$\phi : \mathcal{Y} \to \mathbb{R}^{K+1}, \quad \phi(y) = (y^0, y^1, \cdots, y^K)$$

*and let $\psi$ be an interpolating, continuous positive-definite kernel. Define*

$$\mathcal{H}_M = \left\{ \sum_{i=1}^M \phi(y_i)\psi(\,\cdot\,, \boldsymbol{x}_i) : (\boldsymbol{x}_i, y_i)_{i=1}^M \subseteq \mathcal{Z}_M' \right\} \subseteq \mathcal{H}^{K+1}, \tag{5}$$

*where $\mathcal{H}^{K+1} = \mathcal{H} \times \cdots \times \mathcal{H}$ is the $(K+1)$-dimensional-vector–valued–function Hilbert space constructed from the RKHS $\mathcal{H}$ for which $\psi$ is a reproducing kernel and endowed with the inner product $\langle f, g \rangle_{\mathcal{H}^{K+1}} = \sum_{i=1}^{K+1} \langle f_i, g_i \rangle_{\mathcal{H}}$. Then the embedding*

$$E_M : [\mathcal{Z}_M'] \to \mathcal{H}_M, \quad E_M([(\boldsymbol{x}_1, y_1), \ldots, (\boldsymbol{x}_M, y_M)]) = \sum_{i=1}^M \phi(y_i)\psi(\,\cdot\,, \boldsymbol{x}_i)$$

*is injective, hence invertible, and continuous.*

*Proof.* First, we show that $E_M$ is injective. Suppose that

$$\sum_{i=1}^M \phi(y_i)\psi(\,\cdot\,, \boldsymbol{x}_i) = \sum_{i=1}^M \phi(y_i')\psi(\,\cdot\,, \boldsymbol{x}_i').$$

Denote $\boldsymbol{X} = (\boldsymbol{x}_1, \ldots, \boldsymbol{x}_M)$ and $\boldsymbol{y} = (y_1, \ldots, y_M)$, and denote $\boldsymbol{X}'$ and $\boldsymbol{y}'$ similarly. Taking the inner product with any $f \in \mathcal{H}$ on both sides and using the reproducing property of $\psi$, this implies that

$$\sum_{i=1}^M \phi(y_i)f(\boldsymbol{x}_i) = \sum_{i=1}^M \phi(y_i')f(\boldsymbol{x}_i')$$

for all $f \in \mathcal{H}$. In particular, since by construction $\phi_1(\,\cdot\,) = 1$,

$$\sum_{i=1}^M f(\boldsymbol{x}_i) = \sum_{i=1}^M f(\boldsymbol{x}_i')$$

for all $f \in \mathcal{H}$. Using that $\mathcal{H}$ is interpolating, choose a particular $\hat{\boldsymbol{x}} \in \boldsymbol{X} \cup \boldsymbol{X}'$, and let $f \in \mathcal{H}$ be such that $f(\hat{\boldsymbol{x}}) = 1$ and $f(\,\cdot\,) = 0$ at all other $\boldsymbol{x}_i$ and $\boldsymbol{x}_i'$. Then

$$\sum_{i:\boldsymbol{x}_i=\hat{\boldsymbol{x}}} 1 = \sum_{i:\boldsymbol{x}_i'=\hat{\boldsymbol{x}}} 1,$$

so the number of such $\hat{\boldsymbol{x}}$ in $\boldsymbol{X}$ and the number of such $\hat{\boldsymbol{x}}$ in $\boldsymbol{X}'$ are the same. Since this holds for every $\hat{\boldsymbol{x}}$, $\boldsymbol{X}$ is a permutation of $\boldsymbol{X}'$: $\boldsymbol{X} = \pi(\boldsymbol{X}')$ for some permutation $\pi \in \mathbb{S}_M$. Plugging in the permutation, we can write

$$\sum_{i=1}^M \phi(y_i)f(\boldsymbol{x}_i) = \sum_{i=1}^M \phi(y_i')f(\boldsymbol{x}_i') \overset{(\boldsymbol{X}'=\pi^{-1}(\boldsymbol{X}))}{=} \sum_{i=1}^M \phi(y_i')f(\boldsymbol{x}_{\pi^{-1}(i)}) \overset{(i \leftarrow \pi^{-1}(i))}{=} \sum_{i=1}^M \phi(y_{\pi(i)}')f(\boldsymbol{x}_i).$$

Then, by a similar argument, for any particular $\hat{\boldsymbol{x}}$,

$$\sum_{i:\boldsymbol{x}_i=\hat{\boldsymbol{x}}} \phi(y_i) = \sum_{i:\boldsymbol{x}_i=\hat{\boldsymbol{x}}} \phi(y_{\pi(i)}').$$

Let the number of terms in each sum equal $S$. Since $\mathcal{Z}'_M$ has multiplicity $K$, $S \leq K$. By Lemma 4 from Zaheer et al. (2017), the 'sum-of-power mapping' from $\{y_i : \boldsymbol{x}_i = \hat{\boldsymbol{x}}\}$ to the first $S + 1$ elements of $\sum_{i:\boldsymbol{x}_i=\hat{\boldsymbol{x}}} \phi(y_i)$, i.e. $\left(\sum_{i:\boldsymbol{x}_i=\hat{\boldsymbol{x}}} y_i^0, \dots, \sum_{i:\boldsymbol{x}_i=\hat{\boldsymbol{x}}} y_i^S\right)$, is injective. Therefore,

$$(y_i)_{i:\boldsymbol{x}_i=\hat{\boldsymbol{x}}} \quad \text{is a permutation of} \quad (y'_{\pi(i)})_{i:\boldsymbol{x}_i=\hat{\boldsymbol{x}}}.$$

Note that $\boldsymbol{x}_i = \hat{\boldsymbol{x}}$ for all above $y_i$. Furthermore, note that also $\boldsymbol{x}'_{\pi(i)} = \boldsymbol{x}_i = \hat{\boldsymbol{x}}$ for all above $y'_{\pi(i)}$. We may therefore adjust the permutation $\pi$ such that $y_i = y'_{\pi(i)}$ for all $i$ such that $\boldsymbol{x}_i = \hat{\boldsymbol{x}}$ whilst retaining that $\boldsymbol{x} = \pi(\boldsymbol{x}')$. Performing this adjustment for all $\hat{\boldsymbol{x}}$, we find that $y = \pi(y')$ and $\boldsymbol{x} = \pi(\boldsymbol{x}')$.

Second, we show that $E_M$ is continuous. Compute

$$\left\| \sum_{i=1}^{M} \phi(y_i)\psi(\,\cdot\,, \boldsymbol{x}_i) - \sum_{j=1}^{M} \phi(y'_j)\psi(\,\cdot\,, \boldsymbol{x}'_j) \right\|_{\mathcal{H}^{K+1}}^2$$
$$= \sum_{i=1}^{K+1} \left( \phi_i^\top(\boldsymbol{y})\psi(\boldsymbol{X}, \boldsymbol{X})\phi_i(\boldsymbol{y}) - 2\phi_i^\top(\boldsymbol{y})\psi(\boldsymbol{X}, \boldsymbol{X}')\phi_i(\boldsymbol{y}') + \phi_i^\top(\boldsymbol{y}')\psi(\boldsymbol{X}', \boldsymbol{X}')\phi_i(\boldsymbol{y}') \right),$$

which goes to zero if $[\boldsymbol{X}', \boldsymbol{y}'] \to [\boldsymbol{X}, \boldsymbol{y}]$ by continuity of $\psi$. $\qquad\square$

Having established the injection, we now show that this mapping is a homeomorphism, i.e. that the inverse is continuous. This is formalized in the following lemma.

**Lemma 2.** *Consider Lemma 1. Suppose that $\mathcal{Z}'_M$ is also topologically closed in $\mathcal{A}^M$ and closed under permutations, and that $\psi$ also satisfies (i) $\psi(\boldsymbol{x}, \boldsymbol{x}') \geq 0$, (ii) $\psi(\boldsymbol{x}, \boldsymbol{x}) = \sigma^2 > 0$, and (iii) $\psi(\boldsymbol{x}, \boldsymbol{x}') \to 0$ as $\|\boldsymbol{x}\| \to \infty$. Then $\mathcal{H}_M$ is closed in $\mathcal{H}^{K+1}$ and $E_M^{-1}$ is continuous.*

**Remark 1.** To define $\mathcal{Z}'_2$ with multiplicity one, one might be tempted to define

$$\mathcal{Z}'_2 = \{((\boldsymbol{x}_1, y_1), (\boldsymbol{x}_2, y_2)) \in \mathcal{Z}_2 : \boldsymbol{x}_1 \neq \boldsymbol{x}_2\},$$

which indeed has multiplicity one. Unfortunately, $\mathcal{Z}'_2$ is not closed: if $[0, 1] \subseteq \mathcal{X}$ and $[0, 2] \subseteq \mathcal{Y}$, then $((0, 1), (1/n, 2))_{n=1}^{\infty} \subseteq \mathcal{Z}'_2$, but $((0, 1), (1/n, 2)) \to ((0, 1), (0, 2)) \notin \mathcal{Z}'_2$, because $0$ then has two observations $1$ and $2$. To get around this issue, one can require an arbitrarily small, but non-zero spacing $\epsilon > 0$ between input locations:

$$\mathcal{Z}'_{2,\epsilon} = \{((\boldsymbol{x}_1, y_1), (\boldsymbol{x}_2, y_2)) \in \mathcal{Z}_2 : \|\boldsymbol{x}_1 - \boldsymbol{x}_2\| \geq \epsilon\}.$$

This construction can be generalized to higher numbers of observations and multiplicities as follows:

$$\mathcal{Z}'_{M,K,\epsilon} = \{(\boldsymbol{x}_{\pi(i)}, y_{\pi(i)})_{i=1}^{M} \in \mathcal{Z}_M : \|\boldsymbol{x}_i - \boldsymbol{x}_j\| \geq \epsilon \text{ for } i, j \in [K], \pi \in \mathbb{S}_M\}.$$

**Remark 2.** Before moving on to the proof of Lemma 2, we remark that Lemma 2 would directly follow if $\mathcal{Z}'_M$ were bounded: then $\mathcal{Z}'_M$ is compact, so $E_M$ is a continuous, invertible map between a compact space and a Hausdorff space, which means that $E_M^{-1}$ must be continuous. The intuition that the result must hold for unbounded $\mathcal{Z}'_M$ is as follows. Since $\phi_1(\,\cdot\,) = 1$, for every $f \in \mathcal{H}_M$, $f_1$ is a summation of $M$ "bumps" (imagine the EQ kernel) of the form $\psi(\,\cdot\,, \boldsymbol{x}_i)$ placed throughout $\mathcal{X}$. If one of these bumps goes off to infinity, then the function cannot uniformly converge pointwise, which means that the function cannot converge in $\mathcal{H}$ (if $\psi$ is sufficiently nice). Therefore, if the function does converge in $\mathcal{H}$, $(\boldsymbol{x}_i)_{i=1}^{M}$ must be bounded, which brings us to the compact case. What makes this work is the *density channel* $\phi_1(\,\cdot\,) = 1$, which forces $(\boldsymbol{x}_i)_{i=1}^{M}$ to be well behaved. The above argument is formalized in the proof of Lemma 2.

*Proof.* Define

$$\mathcal{Z}_J = ([-J, J]^d \times \mathcal{Y})^M \cap \mathcal{Z}'_M,$$

which is compact in $\mathcal{A}^M$ as a closed subset of the compact set $([-J, J]^d \times \mathcal{Y})^M$. We aim to show that $\mathcal{H}_M$ is closed in $\mathcal{H}^{K+1}$ and $E^{-1}$ is continuous. To this end, consider a convergent sequence

$$f^{(n)} = \sum_{i=1}^{M} \phi(y_i^{(n)})\psi(\,\cdot\,, \boldsymbol{x}_i^{(n)}) \to f \in \mathcal{H}^{K+1}.$$

Denote $\boldsymbol{X}^{(n)} = (\boldsymbol{x}_1^{(n)}, \ldots, \boldsymbol{x}_M^{(n)})$ and $\boldsymbol{y}^{(n)} = (y_1^{(n)}, \ldots, y_M^{(n)})$. Claim: $(\boldsymbol{X}^{(n)})_{n=1}^{\infty}$ is a bounded sequence, so $(\boldsymbol{X}^{(n)})_{n=1}^{\infty} \subseteq [-J, J]^{dM}$ for $J$ large enough, which means that $(\boldsymbol{X}^{(n)}, \boldsymbol{y}^{(n)})_{n=1}^{\infty} \subseteq \mathcal{Z}_J$ where $\mathcal{Z}_J$ is compact. Note that $[\mathcal{Z}_J]$ is compact in $\mathcal{A}^M/\mathbb{S}_M$ by continuity of the canonical map.

First, we demonstrate that, assuming the claim, $\mathcal{H}_M$ is closed. Note that by boundedness of $(\boldsymbol{X}^{(n)}, \boldsymbol{y}^{(n)})_{n=1}^{\infty}$, $(f^{(n)})_{n=1}^{\infty}$ is in the image of $E_M|_{[\mathcal{Z}_J]} \colon [\mathcal{Z}_J] \to \mathcal{H}_M$. By continuity of $E_M|_{[\mathcal{Z}_J]}$ and compactness of $[\mathcal{Z}_J]$, the image of $E_M|_{[\mathcal{Z}_J]}$ is compact and therefore closed, since every compact subset of a metric space is closed. Therefore, the image of $E_M|_{[\mathcal{Z}_J]}$ contains the limit $f$. Since the image of $E_M|_{[\mathcal{Z}_J]}$ is included in $\mathcal{H}_M$, we have that $f \in \mathcal{H}_M$, which shows that $\mathcal{H}_M$ is closed.

Next, we prove that, assuming the claim, $E_M^{-1}$ is continuous. Consider $E_M|_{[\mathcal{Z}_J]} \colon [\mathcal{Z}_J] \to E_M([\mathcal{Z}_J])$ restricted to its image. Then $(E_M|_{[\mathcal{Z}_J]})^{-1}$ is continuous, because a continuous bijection from a compact space to a metric space is a homeomorphism. Therefore

$$E_M^{-1}(f^{(n)}) = (\boldsymbol{X}^{(n)}, \boldsymbol{y}^{(n)}) = (E_M|_{[\mathcal{Z}_J]})^{-1}(f^{(n)}) \to (E_M|_{[\mathcal{Z}_J]})^{-1}(f) = (\boldsymbol{X}, \boldsymbol{y}).$$

By continuity and invertibility of $E_M$, then $f^{(n)} \to E_M(\boldsymbol{X}, \boldsymbol{y})$, so $E_M(\boldsymbol{X}, \boldsymbol{y}) = f$ by uniqueness of limits. We conclude that $E_M^{-1}(f^{(n)}) \to E_M^{-1}(f)$, which means that $E_M^{-1}$ is continuous.

It remains to show the claim. Let $f_1$ denote the first element of $f$, i.e. the density channel. Using the reproducing property of $\psi$,

$$|f_1^{(n)}(\boldsymbol{x}) - f_1(\boldsymbol{x})| = |\langle \psi(\boldsymbol{x}, \cdot), f_1^{(n)} - f_1 \rangle| \leq \|\psi(\boldsymbol{x}, \cdot)\|_{\mathcal{H}} \|f_1^{(n)} - f_1\|_{\mathcal{H}} = \sigma \|f_1^{(n)} - f_1\|_{\mathcal{H}},$$

so $f_1^{(n)} \to f_1$ in $\mathcal{H}$ means that it does so uniformly pointwise (over $\boldsymbol{x}$). Hence, we can let $N \in \mathbb{N}$ be such that $n \geq N$ implies that $|f_1^{(n)}(\boldsymbol{x}) - f_1(\boldsymbol{x})| < \frac{1}{3}\sigma^2$ for all $\boldsymbol{x}$. Let $R$ be such that $|\psi(\boldsymbol{x}, \boldsymbol{x}_i^{(N)})| < \frac{1}{3}\sigma^2/M$ for $\|\boldsymbol{x}\| \geq R$ and all $i \in [M]$. Then, for $\|\boldsymbol{x}\| \geq R$,

$$|f_1^{(N)}(\boldsymbol{x})| \leq \sum_{i=1}^{M} |\psi(\boldsymbol{x}, \boldsymbol{x}_i^{(N)})| < \frac{1}{3}\sigma^2 \implies |f_1(\boldsymbol{x})| \leq |f_1^{(N)}(\boldsymbol{x})| + |f_1^{(N)}(\boldsymbol{x}) - f_1(\boldsymbol{x})| < \frac{2}{3}\sigma^2.$$

At the same time, by pointwise non-negativity of $\psi$, we have that

$$f_1^{(n)}(\boldsymbol{x}_i^{(n)}) = \sum_{j=1}^{M} \psi(\boldsymbol{x}_j^{(n)}, \boldsymbol{x}_i^{(n)}) \geq \psi(\boldsymbol{x}_i^{(n)}, \boldsymbol{x}_i^{(n)}) = \sigma^2.$$

Towards contradiction, suppose that $(\boldsymbol{X}^{(n)})_{n=1}^{\infty}$ is unbounded. Then $(\boldsymbol{x}_i^{(n)})_{n=1}^{\infty}$ is unbounded for some $i \in [M]$. Therefore, $\|\boldsymbol{x}_i^{(n)}\| \geq R$ for some $n \geq N$, so

$$\frac{2}{3}\sigma^2 > |f_1(\boldsymbol{x}_i^{(n)})| \geq |f_1^{(n)}(\boldsymbol{x}_i^{(n)})| - |f_1^{(n)}(\boldsymbol{x}_i^{(n)}) - f_1(\boldsymbol{x}_i^{(n)})| \geq \sigma^2 - \frac{1}{3}\sigma^2 = \frac{2}{3}\sigma^2,$$

which is a contradiction. $\qquad\square$

The following lemma states that we may construct an encoding for sets containing no more than $M$ elements into a function space, where the encoding is injective and every restriction to a fixed set size is a homeomorphism.

**Lemma 3.** *For every $m \in [M]$, consider a collection $\mathcal{Z}_m' \subseteq \mathcal{Z}_m$ that (i) has multiplicity $K$, (ii) is topologically closed, and (iii) is closed under permutations. Set*

$$\phi : \mathcal{Y} \to \mathbb{R}^{K+1}, \quad \phi(y) = (y^0, y^1, \cdots, y^K)$$

*and let $\psi$ be an interpolating, continuous positive-definite kernel that satisfies (i) $\psi(\boldsymbol{x}, \boldsymbol{x}') \geq 0$, (ii) $\psi(\boldsymbol{x}, \boldsymbol{x}) = \sigma^2 > 0$, and (iii) $\psi(\boldsymbol{x}, \boldsymbol{x}') \to 0$ as $\|\boldsymbol{x}\| \to \infty$. Define*

$$\mathcal{H}_m = \left\{ \sum_{i=1}^{m} \phi(y_i)\psi(\cdot, \boldsymbol{x}_i) : (\boldsymbol{x}_i, y_i)_{i=1}^{m} \subseteq \mathcal{Z}_m' \right\} \subseteq \mathcal{H}^{K+1}, \tag{6}$$

*where $\mathcal{H}^{K+1} = \mathcal{H} \times \cdots \times \mathcal{H}$ is the $(K+1)$-dimensional-vector–valued–function Hilbert space constructed from the RKHS $\mathcal{H}$ for which $\psi$ is a reproducing kernel and endowed with the inner product $\langle f, g \rangle_{\mathcal{H}^{K+1}} = \sum_{i=1}^{K+1} \langle f_i, g_i \rangle_{\mathcal{H}}$. Denote*

$$[\mathcal{Z}_{\leq M}'] = \bigcup_{m=1}^{M} [\mathcal{Z}_m'] \quad \text{and} \quad \mathcal{H}_{\leq M} = \bigcup_{m=1}^{M} \mathcal{H}_m.$$

*Then $(\mathcal{H}_m)_{m=1}^M$ are pairwise disjoint. It follows that the embedding $E$*

$$E \colon [\mathcal{Z}'_{\leq M}] \to \mathcal{H}_{\leq M}, \quad E([Z]) = E_m([Z]) \quad \text{if} \quad [Z] \in [\mathcal{Z}'_m]$$

*is injective, hence invertible. Denote this inverse by $E^{-1}$, where $E^{-1}(f) = E_m^{-1}(f)$ if $f \in \mathcal{H}_m$.*

*Proof.* Recall that $E_m$ is injective for every $m \in [M]$. Hence, to demonstrate that $E$ is injective it remains to show that $(\mathcal{H}_m)_{m=1}^M$ are pairwise disjoint. To this end, suppose that

$$\sum_{i=1}^m \phi(y_i)\psi(\,\cdot\,, \boldsymbol{x}_i) = \sum_{i=1}^{m'} \phi(y'_i)\psi(\,\cdot\,, \boldsymbol{x}'_i)$$

for $m \neq m'$. Then, by arguments like in the proof of Lemma 1,

$$\sum_{i=1}^m \phi(y_i) = \sum_{i=1}^{m'} \phi(y'_i).$$

Since $\phi_1(\,\cdot\,) = 1$, this gives $m = m'$, which is a contradiction. Finally, by repeated application of Lemma 2, $E_m^{-1}$ is continuous for every $m \in [M]$. $\qquad\square$

**Lemma 4.** *Let $\Phi\colon [\mathcal{Z}'_{\leq M}] \to C_b(\mathcal{X}, \mathcal{Y})$ be a map from $[\mathcal{Z}'_{\leq M}]$ to $C_b(\mathcal{X}, \mathcal{Y})$, the space of continuous bounded functions from $\mathcal{X}$ to $\mathcal{Y}$, such that every restriction $\Phi|_{[\mathcal{Z}'_m]}$ is continuous, and let $E$ be from Lemma 3. Then*

$$\Phi \circ E^{-1} \colon \mathcal{H}_{\leq M} \to C_b(\mathcal{X}, \mathcal{Y})$$

*is continuous.*

*Proof.* Recall that, due to Lemma 1, for every $m \in [M]$, $E_m^{-1}$ is continuous and has image $[\mathcal{Z}'_m]$. By the continuity of $\Phi|_{[\mathcal{Z}'_m]}$, then $\Phi|_{[\mathcal{Z}'_m]} \circ E_m^{-1}$ is continuous for every $m \in [M]$. Since $\Phi \circ E^{-1}|_{\mathcal{H}_m} = \Phi|_{[\mathcal{Z}'_m]} \circ E_m^{-1}$ for all $m \in [M]$, we have that $\Phi \circ E^{-1}|_{\mathcal{H}_m}$ is continuous for all $m \in [M]$. Therefore, as $\mathcal{H}_m$ is closed in $\mathcal{H}_{\leq M}$ for every $m \in [M]$, the pasting lemma (Munkres, 1974) yields that $\Phi \circ E^{-1}$ is continuous. $\qquad\square$

From here on, we let $\psi$ be a stationary kernel, which means that it only depends on the difference of its arguments and can be seen as a function $\mathcal{X} \to \mathbb{R}$.

### A.3 PROOF OF THEOREM 1

With the above results in place, we are finally ready to prove our central result, Theorem 1.

**Theorem 1.** For every $m \in [M]$, consider a collection $\mathcal{Z}'_m \subseteq \mathcal{Z}_m$ that (i) has multiplicity $K$, (ii) is topologically closed, (iii) is closed under permutations, and (iv) is closed under translations. Set

$$\phi : \mathcal{Y} \to \mathbb{R}^{K+1}, \quad \phi(y) = (y^0, y^1, \cdots, y^K)$$

and let $\psi$ be an interpolating, continuous positive-definite kernel that satisfies (i) $\psi(\boldsymbol{x}, \boldsymbol{x}') \geq 0$, (ii) $\psi(\boldsymbol{x}, \boldsymbol{x}) = \sigma^2 > 0$, and (iii) $\psi(\boldsymbol{x}, \boldsymbol{x}') \to 0$ as $\|\boldsymbol{x}\| \to \infty$. Define

$$\mathcal{H}_m = \left\{ \sum_{i=1}^m \phi(y_i)\psi(\,\cdot\,, \boldsymbol{x}_i) : (\boldsymbol{x}_i, y_i)_{i=1}^m \subseteq \mathcal{Z}'_m \right\} \subseteq \mathcal{H}^{K+1}, \tag{7}$$

where $\mathcal{H}^{K+1} = \mathcal{H} \times \cdots \times \mathcal{H}$ is the $(K+1)$-dimensional-vector–valued–function Hilbert space constructed from the RKHS $\mathcal{H}$ for which $\psi$ is a reproducing kernel and endowed with the inner product $\langle f, g \rangle_{\mathcal{H}^{K+1}} = \sum_{i=1}^{K+1} \langle f_i, g_i \rangle_{\mathcal{H}}$. Denote

$$\mathcal{Z}'_{\leq M} = \bigcup_{m=1}^M \mathcal{Z}'_m \quad \text{and} \quad \mathcal{H}_{\leq M} = \bigcup_{m=1}^M \mathcal{H}_m.$$

Then a function $\Phi\colon \mathcal{Z}'_{\leq M} \to C_b(\mathcal{X}, \mathcal{Y})$ satisfies (i) continuity of the restriction $\Phi|_{\mathcal{Z}_m}$ for every $m \in [M]$, (ii) permutation invariance (Property 1), and (iii) translation equivariance (Property 2) if and only if it has a representation of the form

$$\Phi(Z) = \rho\left(E(Z)\right), \quad E((\boldsymbol{x}_1, y_1), \ldots, (\boldsymbol{x}_m, y_m)) = \textstyle\sum_{i=1}^{m} \phi(y_i)\psi(\,\cdot\, - \boldsymbol{x}_i)$$

where $\rho\colon \mathcal{H}_{\leq M} \to C_b(\mathcal{X}, \mathcal{Y})$ is continuous and translation equivariant.

*Proof of sufficiency.* To begin with, note that permutation invariance (Property 1) and translation equivariance (Property 2) for $\Phi$ are well defined, because $\mathcal{Z}'_{\leq M}$ is closed under permutations and translations by assumption. First, $\Phi$ is permutation invariant, because addition is commutative and associative. Second, that $\Phi$ is translation equivariant (Property 2) follows from a direct verification and that $\rho$ is also translation equivariant:

$$
\begin{aligned}
\Phi(T_{\boldsymbol{\tau}} Z) &= \rho\left(\sum_{i=1}^{M} \phi(y_i)\psi(\,\cdot\, - (\boldsymbol{x}_i + \boldsymbol{\tau}))\right) \\
&= \rho\left(\sum_{i=1}^{M} \phi(y_i)\psi((\,\cdot\, - \boldsymbol{\tau}) - \boldsymbol{x}_i)\right) \\
&= \rho\left(\sum_{i=1}^{M} \phi(y_i)\psi(\,\cdot\, - \boldsymbol{x}_i)\right)(\,\cdot\, - \boldsymbol{\tau}) \\
&= \Phi(Z)(\,\cdot\, - \boldsymbol{\tau}) \\
&= T'_{\boldsymbol{\tau}}\Phi(Z).
\end{aligned}
$$

$\square$

*Proof of necessity.* Our proof follows the strategy used by Zaheer et al. (2017); Wagstaff et al. (2019). To begin with, since $\Phi$ is permutation invariant (Property 1), we may define

$$\Phi\colon \bigcup_{m=1}^{M} [\mathcal{Z}'_m] \to C_b(\mathcal{X}, \mathcal{Y}), \quad \Phi(Z) = \Phi([Z]),$$

for which we verify that every restriction $\Phi|_{[\mathcal{Z}'_m]}$ is continuous. By invertibility of $E$ from Lemma 3, we have $[Z] = E^{-1}(E([Z]))$. Therefore,

$$\Phi(Z) = \Phi([Z]) = \Phi(E^{-1}(E([Z]))) = (\Phi \circ E^{-1})\left(\sum_{i=1}^{M} \phi(y_i)\psi(\,\cdot\, - \boldsymbol{x}_i)\right).$$

Define $\rho\colon \mathcal{H}_{\leq M} \to C_b(\mathcal{X}, \mathcal{Y})$ by $\rho = \Phi \circ E^{-1}$. First, $\rho$ is continuous by Lemma 4. Second, $E^{-1}$ is translation equivariant, because $\psi$ is stationary. Also, by assumption $\Phi$ is translation equivariant (Property 2). Thus, their composition $\rho$ is also translation equivariant. $\square$

**Remark 3.** The function $\rho\colon \mathcal{H}_{\leq M} \to C_b(\mathcal{X}, \mathcal{Y})$ may be continuously extended to the entirety of $\mathcal{H}^{K+1}$ using a generalisation of the Tietze Extension Theorem by Dugundji et al. (1951). There are variants of Dugundji's Theorem that also preserve translation equivariance.

## B  BASELINE NEURAL PROCESS MODELS

In both our 1d and image experiments, our main comparison is to conditional neural process models. In particular, we compare to a vanilla CNP (1d only; Garnelo et al. (2018a)) and an ATTNCNP (Kim et al., 2019). Our architectures largely follow the details given in the relevant publications.

**CNP baseline.** Our baseline CNP follows the implementation provided by the authors.[6] The encoder is a 3-layer MLP with 128 hidden units in each layer, and RELU non-linearities. The encoder embeds every context point into a representation, and the representations are then averaged across each context set. Target inputs are then concatenated with the latent representations, and passed to the decoder. The decoder follows the same architecture, outputting mean and standard deviation channels for each input.

**Attentive CNP baseline.** The ATTNCNP we use corresponds to the deterministic path of the model described by Kim et al. (2019) for image experiments. Namely, an encoder first embeds each context point $c$ to a latent representation $(\mathbf{x}^{(c)}, \mathbf{y}^{(c)}) \mapsto \mathbf{r}_{xy}^{(c)} \in \mathbb{R}^{128}$. For the image experiments, this is achieved using a 2-hidden layer MLP of hidden dimensions 128. For the 1d experiments, we use the same encoder as the CNP above. Every context point then goes through two stacked self-attention layers. Each self-attention layer is implemented with an 8-headed attention, a skip connection, and two layer normalizations (as described in Parmar et al. (2018), modulo the dropout layer). To predict values at each target point $t$, we embed $\mathbf{x}^{(t)} \mapsto \mathbf{r}_x^{(t)}$ and $\mathbf{x}^{(c)} \mapsto \mathbf{r}_x^{(c)}$ using the same single hidden layer MLP of dimensions 128. A target representation $\mathbf{r}_{xy}^{(t)}$ is then estimated by applying cross-attention (using an 8-headed attention described above) with keys $\mathrm{K} \coloneqq \{\mathbf{r}_x^{(c)}\}_{c=1}^C$, values $\mathrm{V} \coloneqq \{\mathbf{r}_{xy}^{(c)}\}_{c=1}^C$, and query $\mathbf{q} \coloneqq \mathbf{r}_x^{(t)}$. Given the estimated target representation $\hat{\mathbf{r}}_{xy}^{(t)}$, the conditional predictive posterior is given by a Gaussian pdf with diagonal covariance parametrised by $(\boldsymbol{\mu}^{(t)}, \boldsymbol{\sigma}_{\mathrm{pre}}^{(t)}) = \mathrm{decoder}(\mathbf{r}_{xy}^{(t)})$ where $\boldsymbol{\mu}^{(t)}, \boldsymbol{\sigma}_{\mathrm{pre}}^{(t)} \in \mathbb{R}^3$ and $\mathrm{decoder}$ is a 4 hidden layer MLP with 64 hidden units per layer for the images, and the same decoder as the CNP for the 1d experiments.

Following Le et al. (2018), we enforce we set a minimum standard deviation $\boldsymbol{\sigma}_{\mathrm{min}}^{(t)} = [0.1; 0.1; 0.1]$ to avoid infinite log-likelihoods by using the following post-processed standard deviation:

$$\boldsymbol{\sigma}_{\mathrm{post}}^{(t)} = 0.1\boldsymbol{\sigma}_{\mathrm{min}}^{(t)} + (1 - 0.1)\log(1 + \exp(\boldsymbol{\sigma}_{\mathrm{pre}}^{(t)})) \tag{8}$$

---

[6]https://github.com/deepmind/neural-processes

## C  1-DIMENSIONAL EXPERIMENTS

In this section, we give details regarding our experiments for the 1d data. We begin by detailing model architectures, and then provide details for the data generating processes and training procedures. The density at which we evaluate the grid differs from experiment to experiment, and so the values are given in the relevant subsections. In all experiments, the weights are optimized using Adam (Kingma & Ba, 2015) and weight decay of $10^{-5}$ is applied to all model parameters. The learning rates are specified in the following subsections.

### C.1  CNN ARCHITECTURES

Throughout the experiments (Sections 5.1 to 5.3), we consider two models: CONVCNP (which utilizes a smaller architecture), and CONVCNPXL (with a larger architecture). For all architectures, the input kernel $\psi$ was an EQ (exponentiated quadratic) kernel with a learnable length scale parameter, as detailed in Section 4, as was the kernel for the final output layer $\psi_\rho$. When dividing by the density channel, we add $\varepsilon = 10^{-8}$ to avoid numerical issues. The length scales for the EQ kernels are initialized to twice the spacing $1/\gamma^{1/d}$ between the discretization points $(t_i)_{i=1}^T$, where $\gamma$ is the density of these points and $d$ is the dimensionality of input space $\mathcal{X}$.

Moreover, we emphasize that the size of the receptive field is a product of the width of the CNN filters and the spacing between the discretization points. Consequently, for a fixed width kernel of the CNN, as the number of discretization points increases, the receptive field size decreases. One potential improvement that was not employed in our experiments, is the use of depthwise-separable convolutions (Chollet, 2017). These dramatically reduce the number of parameters in a convolutional layer, and can be used to increase the CNN filter widths, thus allowing one to increase the number of discretization points without reducing the receptive field.

The architectures for CONVCNP and CONVCNPXL are described below.

**CONVCNP.** For the 1d experiments, we use a simple, 4-layer convolutional architecture, with RELU nonlinearities. The kernel size of the convolutional layers was chosen to be 5, and all employed a stride of length 1 and zero padding of 2 units. The number of channels per layer was set to $[16, 32, 16, 2]$, where the final channels where then processed by the final, EQ-based layer of $\rho$ as mean and standard deviation channels. We employ a SOFTPLUS nonlinearity on the standard deviation channel to enforce positivity. This model has 6,537 parameters.

**CONVCNPXL.** Our large architecture takes inspiration from UNet (Ronneberger et al., 2015). We employ a 12-layer architecture with skip connections. The number of channels is doubled every layer for the first 6 layers, and halved every layer for the final 6 layers. We use concatenation for the skip connections. The following describes which layers are concatenated, where $L_i \leftarrow [L_j, L_k]$ means that the input to layer $i$ is the concatenation of the activations of layers $j$ and $k$:

- $L_8 \leftarrow [L_5, L_7]$,
- $L_9 \leftarrow [L_4, L_8]$,
- $L_{10} \leftarrow [L_3, L_9]$,
- $L_{11} \leftarrow [L_2, L_{10}]$,
- $L_{12} \leftarrow [L_1, L_{11}]$.

Like for the smaller architecture, we use RELU nonlinearities, kernels of size 5, stride 1, and zero padding for two units on all layers.

### C.2  SYNTHETIC 1D EXPERIMENTAL DETAILS AND ADDITIONAL RESULTS

The kernels used for the Gaussian Processes which generate the data in this experiment are defined as follows:

- EQ:

$$k(x, x') = e^{-\frac{1}{2}\left(\frac{x-x'}{0.25}\right)^2},$$

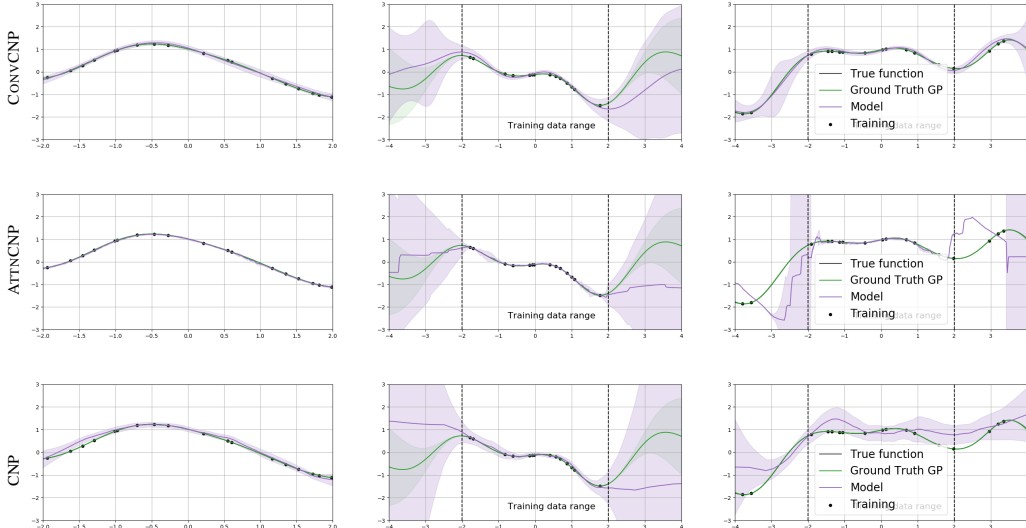

Figure 6: Example functions learned by the (top) CONVCNP, (center) ATTNCNP, and (bottom) CNP when trained on an EQ kernel (with length scale parameter 1). "True function" refers to the sample from the GP prior from which the context and target sets were sub-sampled. "Ground Truth GP" refers to the GP posterior distribution when using the exact kernel and performing posterior inference based on the context set. The left column shows the predictive posterior of the models when data is presented in same range as training. The centre column shows the model predicting outside the training data range when no data is observed there. The right-most column shows the model predictive posteriors when presented with data outside the training data range.

- weakly periodic:

$$k(x, x') = e^{-\frac{1}{2}(f_1(x) - f_1(x'))^2 - \frac{1}{2}(f_2(x) - f_2(x'))^2} \cdot e^{-\frac{1}{8}(x - x')^2},$$

with $f_1(x) = \cos(8\pi x)$ and $f_2(x) = \sin(8\pi x)$, and
- Matern–$\frac{5}{2}$:

$$k(x, x') = (1 + 4\sqrt{5}d + \frac{5}{3}d^2)e^{-\sqrt{5}d}$$

with $d = 4|x - x'|$.

During the training procedure, the number of context points and target points for a training batch are each selected randomly from a uniform distribution over the integers between 3 and 50. This number of context and target points are randomly sampled from a function sampled from the process (a Gaussian process with one of the above kernels or the sawtooth process), where input locations are uniformly sampled from the interval $[-2, 2]$. All models in this experiment were trained for 200 epochs using 256 batches per epoch of batch size 16. We discretize $E(Z)$ by evaluating 64 points per unit in this setting. We use a learning rate of $3\mathrm{e}{-4}$ for all models, except for CONVCNPXL on the sawtooth data, where we use a learning rate of $1\mathrm{e}{-3}$ (this learning rate was too large for the other models).

The random sawtooth samples are generated from the following function:

$$y_{\text{sawtooth}}(t) = \frac{A}{2} - \frac{A}{\pi} \sum_{k=1}^{\infty} (-1)^k \frac{\sin(2\pi k f t)}{k}, \tag{9}$$

where $A$ is the amplitude, $f$ is the frequency, and $t$ is "time". Throughout training, we fix the amplitude to be one. We truncate the series at an integer $K$. At every iteration, we sample a frequency uniformly in $[3, 5]$, $K$ in $[10, 20]$, and a random shift in $[-5, 5]$. As the task is much harder, we sample context and target set sizes over $[3, 100]$. Here the CNP and ATTNCNP employ learning rates of $10^{-3}$. All other hyperparameters remain unchanged.

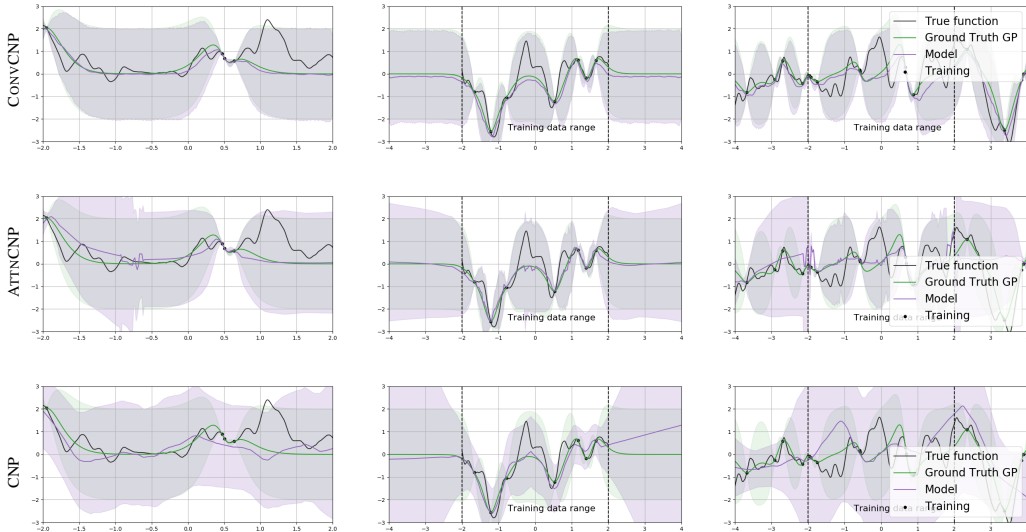

Figure 7: Example functions learned by the (top) CONVCNP, (center) ATTNCNP, and (bottom) CNP when trained on a Matérn-5/2 kernel (with length scale parameter 0.25). "True function" refers to the sample from the GP prior from which the context and target sets were sub-sampled. "Ground Truth GP" refers to the GP posterior distribution when using the exact kernel and performing posterior inference based on the context set. The left column shows the predictive posterior of the models when data is presented in same range as training. The centre column shows the model predicting outside the training data range when no data is observed there. The right-most column shows the model predictive posteriors when presented with data outside the training data range.

| Variable | $m$ | $s$ |
|----------|-----|-----|
| time | $5.94 \times 10^4$ | $8.74 \times 10^2$ |
| lsstu | 1.26 | $1.63 \times 10^2$ |
| lsstg | -0.13 | $3.84 \times 10^2$ |
| lsstr | 3.73 | $3.41 \times 10^2$ |
| lssti | 5.53 | $2.85 \times 10^2$ |
| lsstz | 6.43 | $2.69 \times 10^2$ |
| lssty | 6.27 | $2.93 \times 10^2$ |

Table 4: Values used to normalise the data in the PLAsTiCC experiments.

We include additional figures showing the performance of CONVCNPs, ATTNCNPs and CNPs on GP and sawtooth function regression tasks in Figures 6 to 8.

## C.3 PLASTiCC EXPERIMENTAL DETAILS

The CONVCNP was trained for 200 epochs using 1024 batches of batch size 4 per epoch. For training and testing, the number of context points for a batch are each selected randomly from a uniform distribution over the integers between 1 and the number of points available in the series (usually between 10–30 per bandwidth). The remaining points in the series are used as the target set. For testing, a batch size of 1 was used and statistics were computed over 1000 evaluations. We compare CONVCNP to the GP models used in (Boone, 2019) using the implementation in https://github.com/kboone/avocado. The data used for training and testing is normalized according to $t(v) = (v - m)/s$ with the values in Table 4. These values are estimated from a batch sampled from the training data. To remove outliers in the GP results, log-likelihood values less than $-10$ are removed from the evaluation. These same datapoints were removed from the CONVCNP results as well.

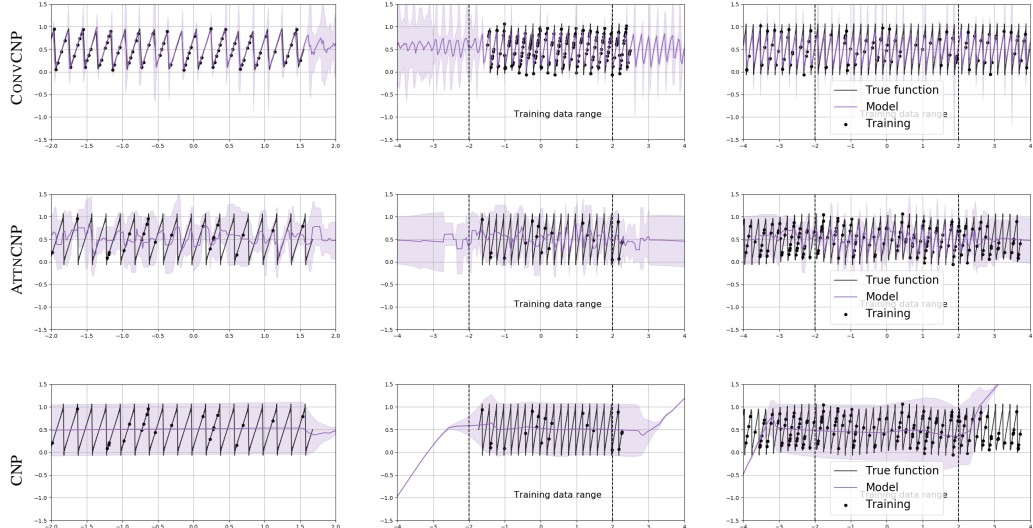

Figure 8: Example functions learned by the (top) CONVCNP, (center) ATTNCNP, and (bottom) CNP when trained on a random sawtooth sample. The left column shows the predictive posterior of the models when data is presented in the same range as training. The centre column shows the model predicting outside the training data range when no data is observed there. The right-most column shows the model predictive posteriors when presented with data outside the training data range.

For this dataset, we only used the CONVCNPXL, as we found the CONVCNP to underfit. The learning rate was set to $10^{-3}$, and we discretize $E(Z)$ by evaluating 256 points per unit.

### C.4 PREDATOR–PREY EXPERIMENTAL DETAILS

We describe the way simulated training data for the experiment in Section 5.3 was generated from the Lotka–Volterra model. The description is borrowed from (Wilkinson, 2011).

Let $X$ be the number of predators and $Y$ the number of prey at any point in our simulation. According to the model, one of the following four events can occur:

$A$: A single predator is born according to rate $\theta_1 XY$, increasing $X$ by one.

$B$: A single predator dies according to rate $\theta_2 X$, decreasing $X$ by one.

$C$: A single prey is born according to rate $\theta_3 Y$, increasing $Y$ by one.

$D$: A single prey dies (is eaten) according to rate $\theta_4 XY$, decreasing $Y$ by one.

The parameter values $\theta_1$, $\theta_2$, $\theta_3$, and $\theta_4$, as well as the initial values of $X$ and $Y$ govern the behavior of the simulation. We choose $\theta_1 = 0.01$, $\theta_2 = 0.5$, $\theta_3 = 1$, and $\theta_4 = 0.01$, which are also used in (Papamakarios & Murray, 2016) and generate reasonable time series. Note that these are likely not the parameter values that would be estimated from the Hudson's Bay lynx–hare data set (Leigh, 1968), but they are used because they yield reasonably oscillating time series. Obtaining oscillating time series from the simulation is sensitive to the choice of parameters and many parametrizations result in populations that simply die out.

Time series are simulated using Gillespie's algorithm (Gillespie, 1977):

1. Draw the time to the next event from an exponential distribution with rate equal to the total rate $\theta_1 XY + \theta_2 X + \theta_3 Y + \theta_4 XY$.

2. Select one of the above events $A$, $B$, $C$, or $D$ at random with probability proportional to its rate.

3. Adjust the appropriate population according to the selected event, and go to 1.

The simulations using these parameter settings can yield a maximum population of approximately 300 while the context set in the lynx–hare data set has an approximate maximum population of about 80 so we scaled our simulation population by a factor of $2/7$. We also remove time series which are longer than 100 units of time, which have more than 10000 events, or where one of the populations is entirely zero. The number of context points $n$ for a training batch are each selected randomly from a uniform distribution between 3 and 80, and the number of target points is $150 - n$. These target and context points are then sampled from the simulated series. The Hudson's Bay lynx–hare data set has time values that range from 1845 to 1935. However, the values supplied to the model range from 0 to 90 to remain consistent with the simulated data.

For evaluation, an interval of 18 points is removed from the the Hudson's Bay lynx–hare data set to act as a target set, while the remaining 72 points act as the context set. This construction highlights the model's interpolation as well as its uncertainty in the presence of missing data.

Models in this setting were trained for 200 epochs with 256 batches per epoch, each batch containing 50 tasks. For this data set, we only used the CONVCNP, as we found the CONVCNPXL to overfit. The learning rate was set to $10^{-3}$, and we discretize $E(Z)$ by evaluating 100 points per unit.

We attempted to train an ATTNCNP for comparison, but due to the nature of the synthetic data generation, many of the training series end before 90 time units, the length of the Hudson's Bay lynx-hare series. Effectively, this means that the ATTNCNP was asked to predict outside of its training interval, a task that it struggles with, as shown in Section 5.1. The plots in Figure 9 show that the ATTNCNP is able to learn the first part of the time series but is unable to model data outside of the first 20 or so time units. Perhaps with more capacity and training epochs the ATTNCNP training would be more successful. Note from Figure 3 that our model does better on the synthetic data than on the real data. This could be due to the parameters of the Lotka–Volterra model used being a poor estimate for the real data.

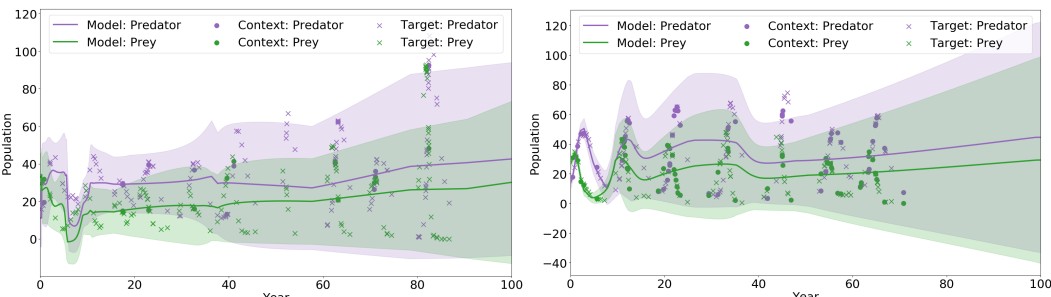

Figure 9: ATTNCNP performance on two samples from the Lotka–Volterra process (sim).

## D  IMAGE EXPERIMENTAL DETAILS AND ADDITIONAL RESULTS

### D.1  EXPERIMENTAL DETAILS

**Training details.** In all experiments, we sample the number of context points uniformly from $\mathcal{U}(\frac{n_{\text{total}}}{100}, \frac{n_{\text{total}}}{2})$, and the number of target points is set to $n_{\text{total}}$. The context and target points are sampled randomly from each of the 16 images per batch. The weights are optimised using Adam (Kingma & Ba, 2015) with learning rate $5 \times 10^{-4}$. We use a maximum of 100 epochs, with early stopping of 15 epochs patience. All pixel values are divided by 255 to rescale them to the $[0, 1]$ range. In the following discussion, we assume that images are RGB, but very similar models can be used for greyscale images or other gridded inputs (e.g. 1d time series sampled at uniform intervals).

**Proposed convolutional CNP.** Unlike ATTNCNP and off-the-grid CONVCNP, on-the-grid CON-VCNP takes advantage of the gridded structure. Namely, the target and context points can be specified in terms of the image, a context mask $M_c$, and a target mask $M_t$ instead of sets of input–value pairs. Although this is an equivalent formulation, it makes it more natural and simpler to implement in standard deep learning libraries. In the following, we dissect the architecture and algorithmic steps succinctly summarized in Section 4. Note that all the convolutional layers are actually depthwise separable (Chollet, 2017); this enables a large kernel size (i.e. receptive fields) while being parameter and computationally efficient.

1. Let I denote the image. Select all context points $\text{signal} := M_c \odot I$ and append a density channel $\text{density} := M_c$, which intuitively says that "there is a point at this position": $[\text{signal}, \text{density}]^\top$. Each pixel value will now have 4 channels: 3 RGB channels and 1 density channel $M_c$. Note that the mask will set the pixel value to 0 at a location where the density channel is 0, indicating there are no points at this position (a missing value).

2. Apply a convolution to the density channel $\text{density}' = \text{CONV}_{\boldsymbol{\theta}}(\text{density})$ and a normalized convolution to the signal $\text{signal}' := \text{CONV}_{\boldsymbol{\theta}}(\text{signal})/\text{density}'$. The normalized convolution makes sure that the output mostly depends on the scale of the signal rather than the number of observed points. The output channel size is 128 dimensional. The kernel size of $\text{CONV}_{\boldsymbol{\theta}}$ depends on the image shape and model used (Table 5). We also enforce element-wise positivity of the trainable filter by taking the absolute value of the kernel weights $\boldsymbol{\theta}$ before applying the convolution. As discussed in Appendix D.4, the normalization and positivity constraints do not empirically lead to improvements for on-the-grid data. Note that in this setting, $E(Z)$ is $[\text{signal}', \text{density}']^\top$.

3. Now we describe the on-the-grid version of $\rho(\cdot)$, which we decompose into two stages. In the first stage, we apply a CNN to $[\text{signal}', \text{density}']^\top$. This CNN is composed of residual blocks (He et al., 2016), each consisting of 1 or 2 (Table 5) convolutional layers with ReLU activations and no batch normalization. The number of output channels in each layer is 128. The kernel size is the same across the whole network, but depends on the image shape and model used (Table 5).

4. In the second stage of $\rho(\cdot)$, we apply a shared pointwise MLP : $\mathbb{R}^{128} \to \mathbb{R}^{2C}$ (we use the same architecture as used for the ATTNCNP decoder) to the output of the first stage at each pixel location in the target set. Here $C$ denotes the number of channels in the image. The first $C$ outputs of the MLP are treated as the means of a Gaussian predictive distribution, and the last $C$ outputs are treated as the standard deviations. These then pass through the positivity-enforcing function shown in Equation (8).

Table 5: CNN architecture for the image experiments.

| Model | Input Shape | $\text{CONV}_{\boldsymbol{\theta}}$ Kernel Size | CNN Kernel Size | CNN Num. Res. Blocks | Conv. Layers per Block |
|---|---|---|---|---|---|
| CONVCNP | < 50 pixels | 9 | 5 | 4 | 1 |
| | > 50 pixels | 7 | 3 | 4 | 1 |
| CONVCNP XL | any | 9 | 11 | 6 | 2 |

## D.2  ZERO SHOT MULTI MNIST (ZSMM) DATA

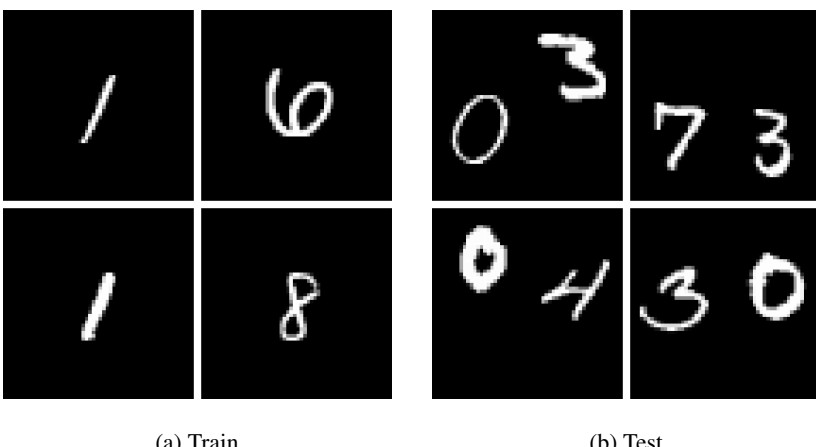

(a) Train                     (b) Test

Figure 10: Samples from our generated Zero Shot Multi MNIST (ZSMM) data set.

In the real world, it is very common to have multiple objects in our field of view which do not interact with each other. Yet, many image data sets in machine learning contain only a single, well-centered object. To evaluate the translation equivariance and generalization capabilities of our model, we introduce the zero-shot multi-MNIST setting.

The training set contains all 60000 $28 \times 28$ MNIST training digits centered on a black $56 \times 56$ background. (Figure 10a). For the test set, we randomly sample with replacement 10000 pairs of digits from the MNIST test set, place them on a black $56 \times 56$ background, and translate the digits in such a way that the digits can be arbitrarily close but cannot overlap (Figure 10b). Importantly, the scale of the digits and the image size are the same during training and testing.

## D.3  ATTNCNP AND CONVCNP QUALITATIVE COMPARISON

Figure 11 shows the test log-likelihood distributions of an ATTNCNP and CONVCNP model as well as some qualitative comparisons between the two.

Although most mean predictions of both models look relatively similar for SVHN and CelebA32, the real advantage of CONVCNP becomes apparent when testing the generalization capacity of both models. Figure 12 shows CONVCNP and ATTNCNP trained on CelebA32 and tested on a downscaled version of Ellen's famous Oscar selfie. We see that CONVCNP generalizes better in this setting. [7]

## D.4  ABLATION STUDY: FIRST LAYER

Table 6: Log-likelihood from image ablation experiments (6 runs).

| Model | MNIST | SVHN | CelebA32 | CelebA64 | ZSMM |
|---|---|---|---|---|---|
| CONVCNP | 1.19 ±0.01 | 3.89 ±0.01 | 3.19 ±0.02 | 3.64 ±0.01 | 1.21 ±0.00 |
| . . . no density | 1.15 ±0.01 | 3.88 ±0.01 | 3.15 ±0.02 | 3.62 ±0.01 | 1.13 ±0.08 |
| . . . no norm. | 1.19 ±0.01 | 3.86 ±0.03 | 3.16 ±0.03 | 3.62 ±0.01 | 1.20 ±0.01 |
| . . . no abs. | 1.15 ±0.02 | 3.83 ±0.02 | 3.08 ±0.03 | 3.56 ±0.01 | 1.15 ±0.01 |
| . . . no abs. norm. | 1.19 ±0.01 | 3.86 ±0.03 | 3.16 ±0.03 | 3.62 ±0.01 | 1.20 ±0.01 |
| . . . EQ | 1.18 ±0.00 | 3.89 ±0.01 | 3.18 ±0.02 | 3.63 ±0.01 | 1.21 ±0.00 |

---

[7]The reconstruction looks worse than Figure 4b despite the larger context set, because the test image has been downscaled and the models are trained on a low resolution CelebA32. These constraints come from ATTNCNP's large memory footprint.

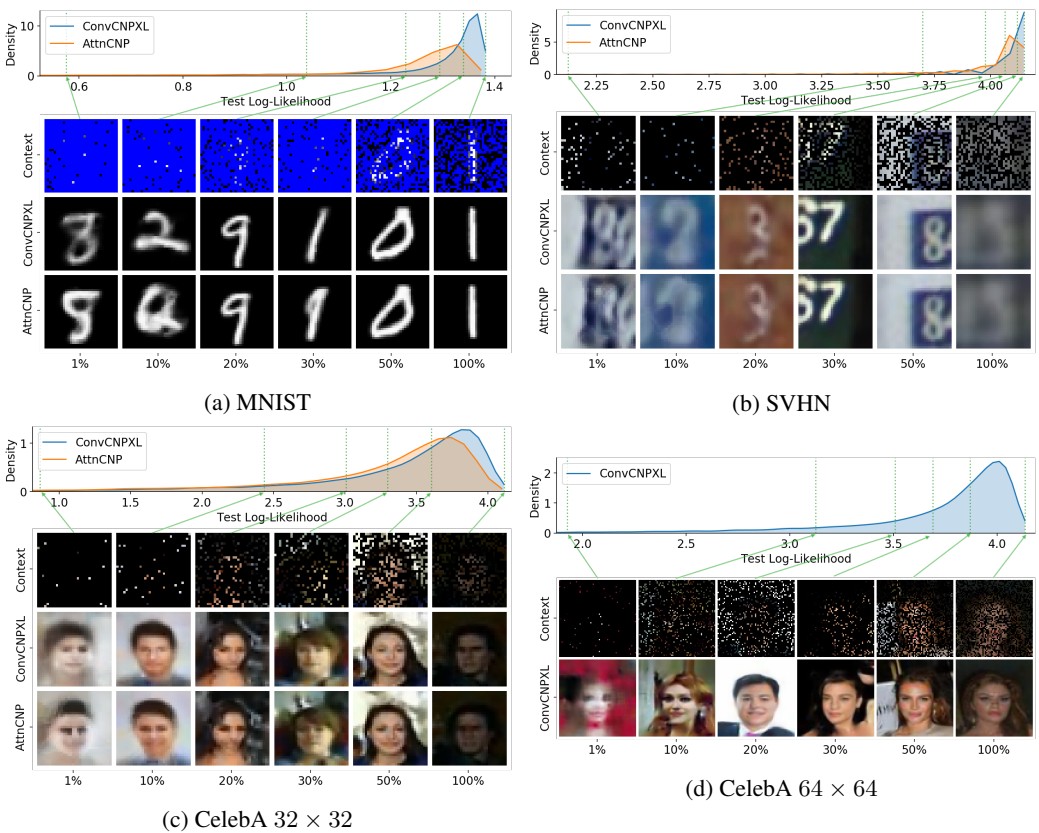

Figure 11: Log-likelihood and qualitative comparisons between ATTNCNP and CONVCNP on four standard benchmarks. The top row shows the log-likelihood distribution for both models. The images below correspond to the context points (top), CONVCNP target predictions (middle), and ATTNCNP target predictions (bottom). Each column corresponds to a given percentile of the CONVCNP distribution. ATTNCNP could not be trained on CelebA64 due to its memory inefficiency.

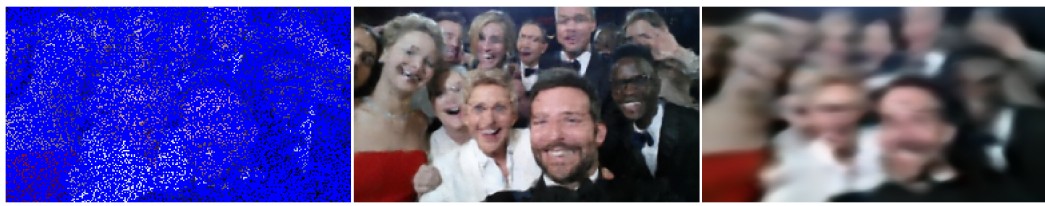

Figure 12: Qualitative evaluation of a CONVCNP (center) and ATTNCNP (right) trained on CelebA32 and tested on a downscaled version ($146 \times 259$) of Ellen's Oscar selfie (DeGeneres, 2014) with $20\%$ of the pixels as context (left).

To understand the importance of the different components of the first layer, we performed an ablation study by removing the density normalization (CONVCNP no norm.), removing the density channel (CONVCNP no dens.), removing the positivity constraints (CONVCNP no abs.), removing the positivity constraints and the normalization (CONVCNP no abs. norm.), and replacing the fully trainable first layer by an EQ kernel similar to the continuous case (CONVCNP EQ).

Table 6 shows the following: (i) Appending a density channel helps. (ii) Enforcing the positivity constraint is only important when using a normalized convolution. (iii) Using a less expressive EQ filter does not significantly decrease performance, suggesting that the model might be learning similar filters (Appendix D.5).

## D.5 Qualitative Analysis of the First Filter

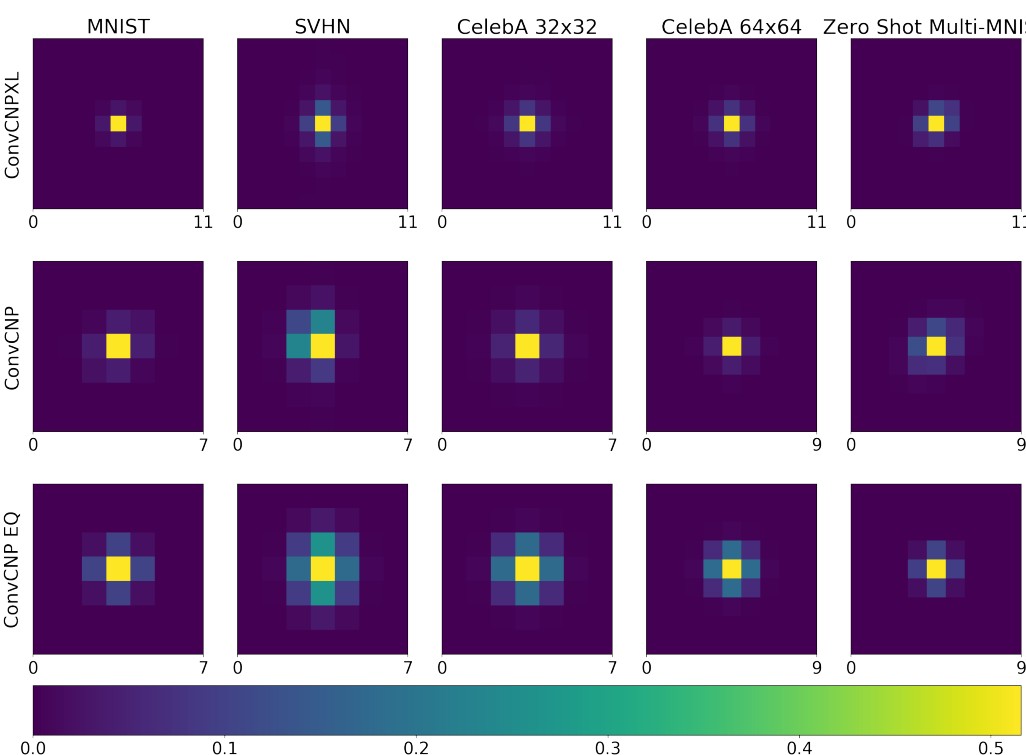

Figure 13: First filter learned by CONVCNPXL, CONVCNP, and CONVCNP EQ for all our datasets. In the case of RGB images, the plotted filters are for the first channel (red). Note that not all filters are of the same size.

As discussed in Appendix D.4, using a less expressive EQ filter does not significantly decrease performance. Figure 13 shows that this happens because the fully trainable kernel learns to approximate the EQ filter.

## D.6 Effect of Receptive Field on Translation Equivariance

As seen in Table 3, a CONVCNPXL with large receptive field performs significantly worse on the ZSMM task than CONVCNP, which has a smaller receptive field. Figure 14 shows a more detailed comparison of the models, and suggests that CONVCNPXL learns to model non-stationary behaviour, namely that digits in the training set are centred. We hypothesize that this issue stems from the the treatment of the image boundaries. Indeed, if the receptive field is large enough and the padding values are significantly different than the inputs to each convolutional layer, the model can learn position-dependent behaviour by "looking" at the distance from the padded boundaries.

For ZSMM, Figure 15 suggests that "circular" padding, where the padding is implied by tiling the image, helps prevent the model from learning non-stationarities, even as the size of the receptive field becomes larger. We hypothesize that this is due to the fact that "circularly" padded values are harder to distinguish from actual values than zeros. We have not tested the effect of padding on other datasets, and note that "circular" padding could result in other issues.

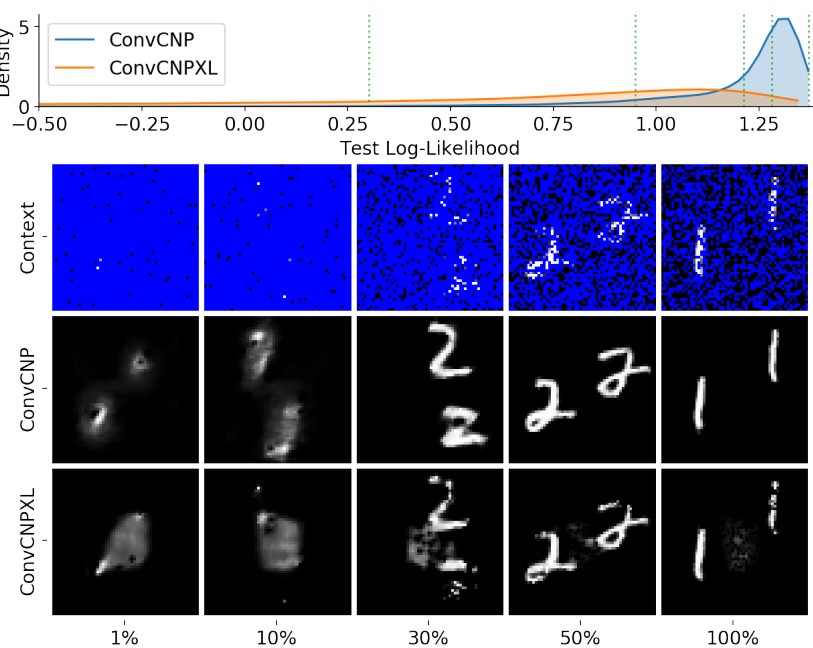

Figure 14: Log-likelihood and qualitative results on ZSMM. The top row shows the log-likelihood distribution for both models. The images below correspond to the context points (top), CONVCNP target predictions (middle), and CONVCNPXL target predictions (bottom). Each column corresponds to a given percentile of the CONVCNP distribution.

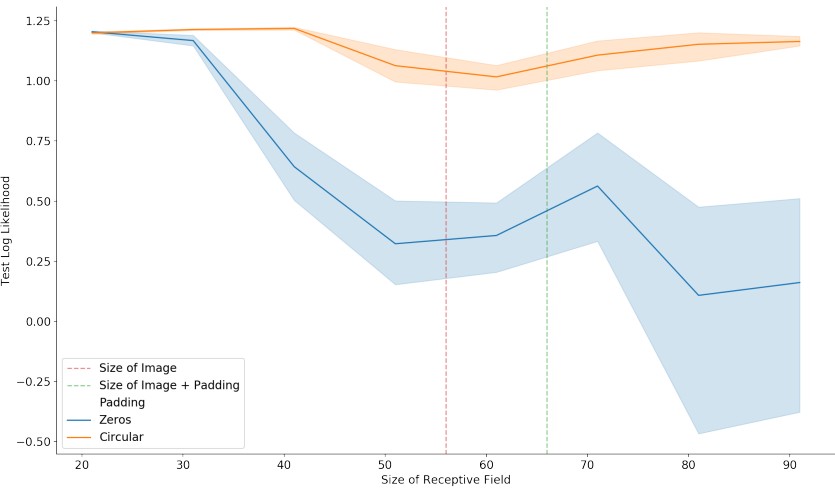

Figure 15: Effect of the receptive field size on ZSMM's log-likelihood. The line plot shows the mean and standard deviation over 6 runs. The blue curve corresponds to a model with zero padding, while the orange one corresponds to "circular" padding.

## E   AUTHOR CONTRIBUTIONS

Richard, Jonathan, and Wessel formulated the project. Richard and Jonathan helped coordinate the team and were closely involved with all aspects of the project. Andrew realised that a density channel was necessary in the model. Yann proposed using normalized convolutions and showed that they led to improvements. All authors contributed to the writing and editing of the paper. Wessel and Jonathan developed the theory. Andrew verified the proof and suggested improvements. James also suggested improvements. Wessel wrote the majority of Appendix A with assistance from Andrew and Jonathan. Andrew and Jonathan performed the initial experiments on simple time-series. Wessel, Jonathan, and James refined these experiments and produced the final versions for the paper. James wrote this section of the paper. James led the experiments on complex time-series and wrote this section of the paper. Andrew worked on the first on-the-grid experiments. Yann redesigned and reimplemented the on-the-grid computational framework and performed all the image experiments shown in the paper. Yann wrote this section of the paper.

