# OpenReview forum: "Convolutional Conditional Neural Processes"
_ICLR.cc/2020/Conference — Accept (Talk)_

### Official Review · AnonReviewer2 · 2019-10-15
**Official Blind Review #2**

**Rating:** 8

**Review:**

The paper introduces ConvCNP, a new member of the neural process(NP) family that models translational equivariance in the data, which uses convolutions and stationary kernels to aggregate the context data into a functional representation.

This problem is well-motivated as there are various domains where such an inductive bias is desirable, such as spatio-temporal data and images, and will help especially with predictions for out-of-distribution tasks. This inductive bias was never built into NPs, and it remained unanswered whether the NP can learn such a behaviour. This paper shows that the answer is negative and that one needs to make modifications to create such inductive bias.

The architecture of the ConvCNP is motivated by theory that completely characterises the set of translation equivariant functions Phi that maps sets of (x,y) pairs to bounded continuous functions that map x to y (disclaimer: I haven’t read through the proof in the appendix, so will not make any claims on its correctness). Theorem 1 defines the set of such functions using rho, phi and psi, and the choices for each on on-the-grid data and off-the-grid data are listed in Section 4. There are ablation studies in Appendix D.4 that justify the choices.

Overall the paper is very well-written and clear for the most part, with helpful pseudo-code and well-laid out quantitative + qualitative results, and a very detailed appendix that allows replicating the setup. The evaluation is extensive, and the results are significant.
    - The results on 1D synthetic data show a noticeable improvement of the ConvCNP compared to the AttnCNP, with improved interpolation as well as accurate extrapolation for the weakly periodic function. I do think however that a more competitive baseline for AttnCNP would have been to parameterise the logits of the attention weights as a periodic function with learnable length scale (e.g. stationary periodic kernel), since this is another way of building in periodicity into the model. Arguably this is more explicit and restrictive than the translational equivariance built into ConvCNP, but would have made for a more interesting comparison.
    - Having said that, I like how the evaluation was performed on a variety of stochastic processes - previous literature only used GP + EQ kernel, but here more challenging non-smooth functions such as GP + Matern kernels and sawtooth functions are explored - and it’s very convincing to see the outstanding performance of ConvCNPs here.
    - It’s also nice to see results on regression tasks on real data (sections 5.2, 5.3), which was never explored in the NP literature as far as I know. 5.2 shows that ConvCNPs can be competitive against other methods that model stochastic processes, and 5.3 shows an instance of where ConvCNPs do a reasonable job whereas (Attn)CNP fails.
    - The results on images is also extensive, covering 6 different datasets (including the 2 zero shot tasks), and show convincing qualitative and quantitative results. The zero shot tasks are nice examples that explicitly show the consequences of not being able to model translation equivariance in more realistic images composed of multiple objects/faces.

I have several comments/questions regarding the disccusion & related work section:
    - One link that might be worth pointing out regarding functional representation of context is that ANP (or AttenCNP) can also be seen as giving a functional representations of the context; the ANP computes a target-specific representation of the context, which can be seen as a function of the target inputs.
    - I think it’s incorrect to say that latent-variable extensions enforce consistency. Even with the latent variable, if the encoder is seen as part of the model, then the NP isn’t consistent (pointed out in the last paragraph of section 2.1 in the ANP paper). So there still are issues regarding AR sampling. There does however seem to exist variants of NPs that satisfy consistency e.g. https://arxiv.org/abs/1906.08324
    - What is preventing the incorporation of a latent variable in the ConvCNP? Is this just something that can be easily done but you haven’t tried, or do you see any non-trivial issues that arise when doing so e.g. maintaining translation equivariance?

Other minor comments:
    - Are there any guidelines on choice of filter size of CNN in the image case? E.g. have you chosen the filter size of ConvCNP such that the receptive field is smaller than the image, whereas it’s bigger for ConvCNPXL? It’s not clear why having a bigger receptive field allows to capture non-stationarity, and it would be helpful to expand on that, perhaps in the appendix.
    - Also it’d help for the sake of clarity to explain why AttnCNP uses significantly more memory than ConvCNP, i.e. because memory for self-attention is O(N^2) where N=HW is the number of inputs, whereas for convolutions it’s O(HW).
    - I think it’d also help to state explicitly in the body that AttnCNP is ANP without the latent path when it is introduced.
    - typos: first paragraph of Section 2: Z_M <- Z_m (twice), finitely <- infinitely, Appendix D.1: separabe <- separable

Overall, I think this is a very strong submission and I vote for its acceptance.

**Experience Assessment:**

I have published one or two papers in this area.

**Review Assessment: Checking Correctness Of Derivations And Theory:**

I did not assess the derivations or theory.

**Review Assessment: Checking Correctness Of Experiments:**

I carefully checked the experiments.

**Review Assessment: Thoroughness In Paper Reading:**

I read the paper thoroughly.

---

> ### Author Response · Authors · 2019-11-11
> **Response to review (part 1 of 2)**
>
>
>
> We greatly thank the reviewer for taking the time to read the paper thoroughly and providing a kind and highly detailed assessment. Your major and minor comments are very helpful and will be used towards improving the quality of our work. Towards the end of the discussion period, we will upload a revised version of the manuscript that will reflect your (and the other reviewers’) comments. As we work on the revised manuscript, please see below our comments on your main concerns.
>
> R2.1: Review
>
> > A more competitive baseline for AttnCNP would have been to parameterise the logits of the attention weights as a periodic function with learnable length scale (e.g. stationary periodic kernel), since this is another way of building in periodicity into the model.
>
> We agree that this is an interesting baseline that likely would have performed better than the standard AttnCNP on the periodic kernel and perhaps the sawtooth function. However, the goal of this comparison is to evaluate the same model on multiple kernels, rather than tailoring an individual model to each kernel. From this perspective, we may similarly have replaced the EQ kernel in the ConvCNP encoder with a periodic kernel. We opted to use the same model for every experiment, demonstrating the flexibility and capacity of these models to capture different data modalities.
>
> R2.2: Comments and Questions
>
> > One link that might be worth pointing out regarding functional representation of context is that ANP (or AttenCNP) can also be seen as giving a functional representations of the context; the ANP computes a target-specific representation of the context, which can be seen as a function of the target inputs.
>
> We agree that, viewed thus, the ANP computes a target-specific representation of the context, which is indeed a function of the target inputs. However, key is that traditional DeepSets – used to define the representation in their models – introduce a finite-dimensional bottleneck, whereas ConvDeepSet produces a representation that is infinite dimensional, removing this bottleneck from the model.
>
> >  I think it’s incorrect to say that latent-variable extensions enforce consistency. Even with the latent variable, if the encoder is seen as part of the model, then the NP isn’t consistent (pointed out in the last paragraph of section 2.1 in the ANP paper). So there still are issues regarding AR sampling. There does however seem to exist variants of NPs that satisfy consistency e.g. https://arxiv.org/abs/1906.08324
>
> Thank you for pointing this out. The discussion on consistency in the initial submission is indeed inaccurate and will be corrected in the revision. What was meant is that the construction is guaranteed to be statistically consistent over the non-context points. In the revision, we will make clear that we are referring to this notion of consistency (conditional consistency). This requires a view of the model where the context points are treated separately, which we agree is uncomfortable. If, instead, the context points are also considered part of the model and handled by AR sampling, then again the resulting distribution does not obey statistical consistency. We agree that developing consistent variants would be an interesting direction for future work. We referenced the conditional BRUNO work in the conclusion, and thank you for pointing us towards the Functional Neural Process (FNP), which indeed is also relevant to the discussion. We will mention FNPs in the discussion and add a reference.
>
> > What is preventing the incorporation of a latent variable in the ConvCNP? Is this just something that can be easily done but you haven’t tried, or do you see any non-trivial issues that arise when doing so e.g. maintaining translation equivariance?
>
> We see no major issues with incorporating a latent variable in the ConvCNP. In fact, we think that this constitutes a highly interesting extension, as there are several ways this could be achieved, and these pose several interesting challenges that need to be addressed. We aim to explore this direction in future work.

---

> > ### Author Response · Authors · 2019-11-11
> > **Response to review (part 2 of 2)**
> >
> >
> >
> > R2.3: Minor Comments
> >
> > > Are there any guidelines on choice of filter size of CNN in the image case? E.g. have you chosen the filter size of ConvCNP such that the receptive field is smaller than the image, whereas it’s bigger for ConvCNPXL? It’s not clear why having a bigger receptive field allows to capture non-stationarity, and it would be helpful to expand on that, perhaps in the appendix.
> >
> > The filter size is an important design choice that indeed warrants discussion in the paper. We will add an appendix with new experiments and discussion about the effect of the receptive field on translation equivariance. In the image experiments, the ConvCNP and ConvCNPXL were chosen such that the former has a smaller receptive field than the input, while the latter has a larger one.
> >
> > Empirically, we found that increasing the receptive field decreases the performance of the model on tasks that are reliant on translation equivariance. We believe this has to do with the behaviour of the model at the boundaries of the images, and in particular, we believe this is an artifact of the 0-padding at the boundaries of the images in the ZSMM experiments. We showcase this issue by adding a plot in the appendix showing the test log likelihood against the size of the receptive field for a $\rho$ which uses “zeros” and “circular” padding. With “zeros” padding, the log likelihood decreases relatively smoothly with an increasing receptive field. For “circular” padding, there seems to be no significant correlation between these two.
> >
> > > Also it’d help for the sake of clarity to explain why AttnCNP uses significantly more memory than ConvCNP, i.e. because memory for self-attention is O(N^2) where N=HW is the number of inputs, whereas for convolutions it’s O(HW).
> >
> > Thank you for pointing this out. We will add the theoretical memory complexity of self attention and convolutions in a revised version of the writing.
> >
> > > I think it’d also help to state explicitly in the body that AttnCNP is ANP without the latent path when it is introduced.
> > > typos: first paragraph of Section 2: Z_M <- Z_m (twice), finitely <- infinitely, Appendix D.1: separabe <- separable
> >
> > Thank you. These points will be addressed in a revised version of the writing.

---

### Official Review · AnonReviewer3 · 2019-10-23
**Official Blind Review #3**

**Rating:** 8

**Review:**

-- Summary

This paper considers the problem of developing neural processes which
are translation-equivariant. The authors derive a necessary and sufficient
functional form that the neural process \Phi function must exhibit
in order to be permutation invariant, continuous and translation
equivariant.

Using the derived functional form, the authors construct a
translation-equivariant neural process, the convolutional conditional
neural process.

Results in several experimental settings are given: 1d synthetic
experiments, an astronomical time-series modelling experiment, a
sim2real experiment, and several image completion experiments.  All
the experiments show performance improvements over the AttnCNP, the
main baseline tested against. In the astronomy setting the authors
test against the winning kaggle entry, against which they get better
log likelihood. The authors give several qualitative experiments,
including image completion tasks from a small number of pixels.

Proofs of all the theorems and full details of all the experiments
are given in the appendix, along with ablations of the model.


-- Review

Overall I found this paper very impressive. It is clear how the theoretical
results motivate the choice of architecture. The fact that Theorem 1
completely characterises the design of all translation-equivariant
neural processes is a remarkable result which precisely specifies the
degrees of freedom available when constructing a convolutional NP.

The implementation gives state of the art results against
the AttnCNP while using fewer parameters on a variety of tasks. The image
completion tasks are impressive.

It seems that the authors close an open question posed in (Zaheer 2017)
regarding how to form embeddings of sets of varying size by embedding
the sets into an RKHS instead of a finite-dimensional space. This in itself
is an interesting idea, and I am interested to see how this embedding method
be applied outside of the CNP framework.

The experimental results are comprehensive and diverse, showing good
performance on both toy examples and more real-world problems. The ablations
and qualitative comparisons in the appendix are helpful in showing where
the ConvCNP outperforms the AttnCNP.

My main criticism of the work is that it's very dense, requiring a few
passes to really grasp the theoretical contribution and the concrete
architecture used in the ConvCNP. I would recommend enlarging figure 1
(b), which is illuminating but quite cluttered due to the small
size. Perhaps the section on multiplicity could be moved to the
appendix to make space as it seems for all real-world datasets the
multiplicity would be equal to 1.


Misc Comments

- It would be good to have a brief discussion of why the ConvCNPPXL performs
very badly on the ZSMM task, while being the best performing method in all
of the other tasks. I couldn't find such a discussion.
- Did the authors try emitting a 36-dimensional joint covariance matrix over the
six-dimensional output in the plasticc experiment?
- In the synthetic experiments, for the EQ and weak periodic kernels it would
be nice to see the `ground truth' log-likelihood given by the actual GP,
just to have some idea of what the upper bound of LL could be.
- In appendix C.2 Figure 6, what is the difference between the `true function' and the
`Ground Truth GP'? I thought the true function was a gp...

**Experience Assessment:**

I do not know much about this area.

**Review Assessment: Checking Correctness Of Derivations And Theory:**

I assessed the sensibility of the derivations and theory.

**Review Assessment: Checking Correctness Of Experiments:**

I assessed the sensibility of the experiments.

**Review Assessment: Thoroughness In Paper Reading:**

I read the paper thoroughly.

---

> ### Author Response · Authors · 2019-11-11
> **Response to review**
>
>
>
> We would like to thank the reviewer for a kind and helpful review and useful comments which we believe will improve the paper. We are pleased that you have recognized the role of Theorem 1 in motivating the work, and the variety of the experiments. We address specific comments raised in the review below. Towards the end of the discussion period, we will upload a revised version of the manuscript that will reflect your (and the other reviewers’) comments. As we work on the revised manuscript, please see below our comments on your main concerns.
>
> R3.1 Major Comment
>
> > My main criticism of the work is that it's very dense, requiring a few passes to really grasp the theoretical contribution and the concrete architecture used in the ConvCNP
>
> We agree that there is a large amount of material to cover in the paper. We are working on rewriting Section 4 on the ConvCNP architecture to make it easier to understand our method. We will also enlarge Fig. 1 b) to make it more readable. The section on multiplicity is important to include in the main body since, without it, we cannot accurately state Theorem 1. However, we have replaced the discussion on multiplicity in the main body with a shorter intuitive description, leaving the mathematical details to an appendix. Additionally, we have put considerable effort into improving the clarity and readability of Appendix A, the most technical part of the paper.
>
> R3.2 Miscellaneous Comments
>
> > It would be good to have a brief discussion of why the ConvCNPPXL performs very badly on the ZSMM task, while being the best performing method in all of the other tasks. I couldn't find such a discussion.
>
> Thank you for your question. Good performance on the ZSMM task requires translation equivariance. In practice, we find that when the receptive field is very large, the model exhibits undesirable behaviours at the boundaries of the image. In particular, we believe that this is an artifact of the 0-padding at the boundaries of the images in the ZSMM experiments. We will add a plot in the appendices showing the test log likelihood for ZSMM against the size of the receptive field for a $\rho$ which uses “zeros” and “circular” padding. With ”zeros” padding, the test log likelihood decreases relatively smoothly with an increasing receptive field. For “circular” padding, there seems to be no significant correlation between these two.  We will also add a discussion to this end to the experimental section.
>
> > Did the authors try emitting a 36-dimensional joint covariance matrix over the six-dimensional output in the plasticc experiment?
>
> This is an interesting suggestion, and is a very natural extension of our work for the multi-output regression setting. However, in this work we only emitted independent Gaussian predictive distributions because this was the simplest setting, and our main concern was to judge if the representational power of deep learning combined with translation equivariance could outperform standard GP regression in this setting.
>
> > In the synthetic experiments, for the EQ and weak periodic kernels it would be nice to see the `ground truth' log-likelihood given by the actual GP, just to have some idea of what the upper bound of LL could be.
>
> We agree that this is an interesting baseline to provide, and will include the ground truth GP log-likelihoods in the revised version of the paper.
>
> > In appendix C.2 Figure 6, what is the difference between the `true function' and the `Ground Truth GP'? I thought the true function was a gp.
>
> The true function is a single sample from the GP prior. This sample is then evaluated at several points to obtain a training set. The ground truth GP refers to the posterior obtained by training a GP that has the same kernel as that used to generate the true function. We will change the wording in the paper to make this more clear.

---

> > ### Comment · AnonReviewer3 · 2019-11-15
> > **--**
> >
> > I have read the rebuttal from the authors and am satisfied with their answers! I will maintain my initial assessment.

---

### Official Review · AnonReviewer1 · 2019-10-25
**Official Blind Review #1**

**Rating:** 6

**Review:**

The paper describes a method for model neural for neural processes considering  translation-equivariant embeddings.

The paper seems to be quite specific topic. Maybe, the author could add more empirical results to it to show the impact on translation-equivariant examples. The theoretical claims seem to be valid. So the question is a bit open what are the applications. The empirical results are also narrow as there is not much other competitive work. The results seem to be increment extension to previous work.

The work looks solid to me, currently I am probably not able to appreciate and judge  relevance to its full extend. I would judge, it is more of interest to view specific people working on this - maybe, the authors could for the final version make this more clear.

The questions that should be more addressed maybe is also the  applications - why is this relevant and how does it improve your specific cases. Why do we want to develop this. State of the art is quite relative if authors come from a quit narrow area which not much papers on the topic and data sets.

One of the main points of the paper did not get clear how does translation-equivariant helps to solve or improve the empirical results. Could you add some examples where this improves results.

I remain ambivalent. It seems to be solid work with not much convincing applications and somewhat incremental. Maybe the authors might address this in their introduction more. The motivation remains unclear to me and hence difficult to judge its potential and impact.



**Experience Assessment:**

I do not know much about this area.

**Review Assessment: Checking Correctness Of Derivations And Theory:**

I assessed the sensibility of the derivations and theory.

**Review Assessment: Checking Correctness Of Experiments:**

I assessed the sensibility of the experiments.

**Review Assessment: Thoroughness In Paper Reading:**

I read the paper at least twice and used my best judgement in assessing the paper.

---

> ### Author Response · Authors · 2019-11-11
> **Response to review (part 1 of 2)**
>
>
>
> Summary of the reviewer’s main concerns:
> 	1. How widely applicable is the method developed in the paper?
> 	2. How important is equivariance as an inductive bias?
> 	3. Do the experiments demonstrate that the method is generally applicable? Is the inductive bias (translation equivariance) empirically beneficial?
>
> Summary of the authors’ response:
> 	1. The methods are widely applicable in real-world applications including time-series, spatial data, and images.
> 	2. Equivariance is hugely important providing large performance gains.
> 	3. The experiments show that the method is useful for time-series modelling, sim2real transfer, and image modelling. They also clearly demonstrate the benefits of translation equivariance.
>
> ----
>
> Detailed Rebuttal:
>
> We thank you for your time and effort in reading and reviewing our paper. Towards the end of the discussion period, we will upload a revised version of the manuscript that will reflect your (and the other reviewers’) comments. As we work on the revised manuscript, please see below our comments on your main concerns. We look forward to your response, and to an ongoing discussion on these points.
>
> R1.1: How Widely Applicable is the Model?
>
> Neural process based models are particularly applicable to settings where a large collection of small-but-related datasets are available, and one wishes to construct powerful models that can efficiently provide inferences for unseen datasets. Examples of such settings are abundant: Image reconstruction, as in Section 5.4 of our paper (also featured in the experimental sections of [1–3]) is one such example. Further examples are edge imputation on graphs [4], learning of robotic movement primitives [5], and few-shot classification [1,6]. Importantly, neural processes can also model data which is non-uniformly sampled, e.g. medical time-series data [7]. Such data is difficult to model with CNNs and RNNs, which means that applications with data like this have not fully benefited from the power of deep learning. In our work, we consider additional real-word applications (with non-uniformly sampled data) of neural processes such as modelling of astronomical objects (Section 5.2) and predator-prey models in a Sim-2-Real environment (Section 5.3). All of the above are examples of real-world applications of neural processes, highlighting the flexibility and broad applicability of this model class.
>
> R1.2: How Important is Equivariance as an Inductive Bias?
>
> It is difficult to overstate the practical applicability of translation equivariance as an inductive bias. The general success of CNNs may (arguably) be attributed to this inductive bias in large part. As we discuss in the paper, many of the applications of interest for NP-based models may also greatly benefit from this inductive bias. For example, consider time-series-based applications, such as the synthetic data in Section 5.1, astronomical objects (Section 5.2), and predator–prey models (Section 5.3). These sections demonstrate that our work brings the benefits of convolutions to applications with non-uniformly sampled data, which is an open challenge in the ML literature.  Similarly, as is well known from the standard CNN example, image modelling significantly benefits from this inductive bias (Section 5.4). We agree with you that this motivation can be better developed in the paper. We will work on adding this high-level motivation to the introduction in the revised version of the paper, and thank you for raising the issue.

---

> > ### Author Response · Authors · 2019-11-11
> > **Response to review (part 2 of 2)**
> >
> >
> >
> > R1.3: Scope of the Experiments? Benefit of Translation Equivariance?
> >
> > Next, you mention concerns regarding our experiments, in particular their scope and the lack of specific examples highlighting the usefulness of our model. On this matter, we respectfully disagree, and would like to highlight the following as evidence. First, as noted by both Reviewers 2 and 3, our experimental section is
> >
> > 	- R3: “comprehensive and diverse, showing good performance on both toy examples and more real-world problems”, and
> > 	- R2:  “the evaluation is extensive, and the results are significant”.
> >
> > Further, the empirical evaluation clearly demonstrates the benefits arising directly from translation equivariance. In all of our experiments, the introduction of translation equivariance as an inductive bias results in significant gains, which manifests itself in several ways.
> >
> > 	1. Performance: As pointed out by both Reviewers 2 and 3, on standard performance metrics (e.g., log-likelihood and RMSE), our models achieve significant improvements over powerful but non-translation-equivariant competitors.
> > 	2. Model size: As pointed out by Reviewer 3, our models are (in most cases) far more parameter efficient than their non-translation-equivariant competitors.
> > 	3. Generalization to out-of-distribution data: As pointed out by Reviewer 2, arguably the most convincing empirical demonstration of the usefulness of our model is its ability to generalize to out-of-distribution data. Examples:
> >
> > 		a. Consider Figures 2, 6, 7, and 8. Our model is able to produce high-quality predictive distributions even when encountering data that is out of the training distribution range. We emphasize that this is a direct consequence of translation equivariance, and is therefore something that the non-translation-equivariant baselines are incapable of, as is demonstrated in those same figures.
> > 		b. Consider Figure 4. Our model is able to generalize to images that are significantly different from the training distributions, e.g. containing multiple digits as opposed to a single, centered digit, or images of different shapes containing multiple faces as opposed to a single face. Again, we stress that this is a direct consequence of translation equivariance. Observe that in Figures 4.a and 12 it is apparent that non-translation-equivariant models are incapable of this kind of generalization.
> >
> > As pointed out by both Reviewers 2 and 3, the inductive bias introduced by translation equivariance provides strong motivation for our developments, and the comprehensive 	empirical results corroborate the motivation. We hope our comments address your concerns. We look forward to reading your response, and a continued discussion on these points.
> >
> > [1] M. Garnelo, D. Rosenbaum, C. Maddison, T. Ramalho, D. Saxton, M. Shanahan, Y. W. Teh, D. Rezende, and S. M. A. Eslami. Conditional neural processes. 2018.
> > [2] M. Garnelo, J. Schwarz, D. Rosenbaum, F. Viola, D. J Rezende, S. M. A. Eslami, and Y. W. Teh. Neural processes. 2018.
> > [3] H. Kim, A. Mnih, J. Schwarz, M. Garnelo, S. M. A. Eslami, D. Rosenbaum, O. Vinyals, and Y. W. Teh. Attentive neural processes. 2019.
> > [4] A. Carr, and D. Wingate. Graph neural processes: towards Bayesian graph neural networks. 2019.
> > [5] M. Y. Seker, M. Imre, J. Piater, E. Ugur. Conditional neural movement primitives. 2019.
> > [6] J. Requeima, J. Gordon, J. Bronskill, S. Nowozin, and R. E. Turner. Fast and flexible multi-task classification using conditional neural adaptive processes. 2019.
> > [7] V. Fortuin, M. Huser, F. Locatello, H. Strathmann, and G. Ratsch.  SOM-VAE: Interpretable discrete representation learning on time-series. 2018.

---

### Author Response · Authors · 2019-11-14
**Response to reviews and revised manuscript**


We thank the reviewers for their detailed reviews, and many helpful comments. We have now uploaded a revised version of the manuscript, reflecting the suggestions. The main revisions are summarized below:
	1. We have put significant effort into improving the clarity and readability of the paper, which we realize covers a large amount of material. To improve exposition we have
		a. rewritten large parts of section 4, to include more details on the models;
		b. included pseudo-code for on-the-grid ConvCNP as well;
		c. redesigned Fig 1b to improve readability (now Fig 1a); and
		d. put significant effort into rewriting Appendix A, which is the most technical part of the paper. We believe the proofs are now far clearer and easier to follow.
	2. We have put effort into further explaining the results. In particular, we have focused on the image setting and performance on the ZSMM task, which leads to a discussion on the  relationship between the size of the receptive field and generalization performance. This discussion further provides some insights into designing architectures for the model. This modification includes an expanded discussion in Section 5.4 and new results from an empirical investigation in Appendix D.6.
	3. We have improved the discussion regarding consistency in Section 6, making clear the distinction between consistency and _conditional_ consistency, and including some missing citations.
We believe these changes have improved the quality of the paper, and will lead to greater impact. We thank the reviewers for their useful feedback.

---

### Decision · Program_Chairs · 2019-12-19

**Decision:**

Accept (Talk)

**Comment:**

This paper presents Convolutional Conditional Neural Process (ConvCNP), a new member of the neural process family that models translation equivariance. Current models must learn translation equivariance from the data, and the authors show that ConvCNP can learn this as part of the model, which is much more generalisable and efficient. They evaluate the ConvCNP on several benchmarks, including an astronomical time-series modelling experiment, a sim2real experiment, and several image completion experiments and show excellent results. The authors wrote extensive responses the the reviewers, uploading a revised version of the paper, and there was some further discussion. This is a strong paper worthy of inclusion in ICLR and could have a large impact on many fields in ML/AI.